# Generative Trajectory Stitching through Diffusion Composition

**Yunhao Luo**[1]  **Utkarsh A. Mishra**[1]  **Yilun Du**[2,†]  **Danfei Xu**[1,†]

[1] Georgia Tech  [2] Harvard University

## Abstract

Effective trajectory stitching for long-horizon planning is a significant challenge in robotic decision-making. While diffusion models have shown promise in planning, they are limited to solving tasks similar to those seen in their training data. We propose CompDiffuser, a novel generative approach that can solve new tasks by learning to compositionally stitch together shorter trajectory chunks from previously seen tasks. Our key insight is modeling the trajectory distribution by subdividing it into overlapping chunks and learning their conditional relationships through a single bidirectional diffusion model. This allows information to propagate between segments during generation, ensuring physically consistent connections. We conduct experiments on benchmark tasks of various difficulties, covering different environment sizes, agent state dimension, trajectory types, training data quality, and show that CompDiffuser significantly outperforms existing methods. Project website at https://comp-diffuser.github.io/.

## 1 Introduction

Generative models have demonstrated remarkable capabilities in modeling complex distributions across domains like images, videos, and 3D shapes. In robot planning, these models offer a promising approach by modeling distributions over plan sequences, which allows amortizing the computational cost of traditional search and optimization methods. This effectively transforms planning into sampling likely solutions given start and goal conditions. Recent works like Diffuser [26] and Decision Diffuser [1] have shown how diffusion models can learn to generate entire plans for long-horizon robotics tasks. However, exhaustively modeling joint distributions over entire plan sequences for all possible start and goal states remains extremely sample-inefficient, as it requires collecting long-horizon plan data covering all possible combinations of initial states and goals.

The concept of trajectory stitching [88] from Reinforcement Learning literature [62] presents a potential solution by combining chunks of different trajectories to create new, potentially better policies. The methods work by identifying high-reward trajectory chunks and stitching them together at states where they overlap or are similar enough, creating composite trajectories that can inform better policy learning. This effectively enables compositional generalization since collecting long consecutive trajectories is costly, and these short chunks can be flexibly assembled to complete new tasks. The key challenge lies in finding appropriate stitching points where trajectories can be combined while maintaining dynamic consistency and feasibility. Our goal is to enable generative planners to solve long-horizon tasks without requiring long-horizon training data, while retaining their ability to generate physically feasible, goal-directed plans.

We propose a novel diffusion-based approach, Compositional Diffuser (CompDiffuser), that enables effective trajectory stitching through goal-conditioned causal trajectory generation.

---

[†]Equal advising.

39th Conference on Neural Information Processing Systems (NeurIPS 2025).

Our key insight is that we can model the trajectory distribution compositionally by subdividing it into distributions of overlapping chunks and learning their conditional relationships. Rather than learning separate models for each chunk, we train a single diffusion model that can generate trajectory chunks conditioned on neighboring chunks' states (Figure 1). This allows information to propagate bidirectionally during the reverse diffusion process as illustrated in Figure 2: each chunk's generation is influenced by both past and future chunks. This architecture naturally enables both parallel generation of chunks and causal autoregressive generation, each with different trade-offs in computational cost and planning quality.

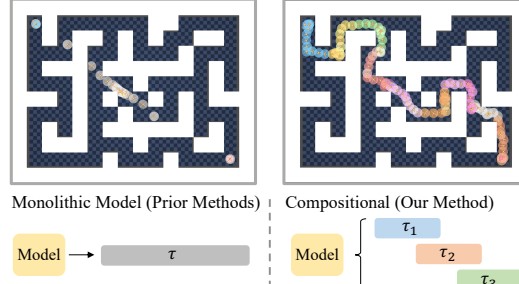

Figure 1: **Compositional Trajectory Generation**. CompDiffuser enables generative trajectory stitching through diffusion composition. Left: Monolithic generative planner fails to generalize to tasks of longer horizon and collapses to the maze center. Right: Our method successfully navigates the ant agent from start to goal by compositionally stitching together shorter trajectories.

We conduct extensive experiments across benchmark tasks of varying difficulty levels, including different environment sizes (from simple U-mazes to complex giant mazes), agent state dimensions (from 2D point agents to 50D humanoid robots), trajectory types (from maze navigation trajectories to ball dribbling trajectories), and training data quality (from clean demonstrations to noisy exploration data). Our results demonstrate that CompDiffuser significantly outperforms multiple imitation learning and offline reinforcement learning baselines across all settings. We show that our approach can effectively solve long-horizon tasks while maintaining plan feasibility and goal-reaching behavior. We validate the importance of our key technical components including the bidirectional conditioning mechanism, the autoregressive sampling process, and the flexible replanning capability.

In summary, the key contributions of this work are:

- A noisy-sample conditioned diffusion planning framework that enables learning compositional trajectory distributions by decomposing the trajectory generation procedure into a sequence of segments each generated by a separate diffusion denoising process.

- A compositional goal-conditioned trajectory planning method that uses bidirectional information propagation during denoising to maintain physical consistency between trajectory chunks.

- A set of empirical results showing significant improvements over existing methods across multiple trajectory stitching benchmarks, with detailed analysis of model capabilities and limitations.

## 2 Related Work

**Diffusion Models For Planning.** Many works have studied the applications of diffusion models [58, 24] for generative planning [26, 1, 54, 72, 22, 64, 42, 86, 7]. Diffusion planning has been widely applied in various fields, such as motion planning [5, 43], procedure planning [67], task planning [76, 16], autonomous driving [75, 37, 68], reasoning [79], and reward learning [49]. Many techniques have also been combined with diffusion planning, including hierarchical planning [35, 8, 45], self-evolving planner [36], preference alignment [10], tree branch-pruning [17], refining [31], replanning [84], uncertainty-aware planning [60], equivariance [4]. However, these works are usually constrained to plan within similar horizons as training data. Our work instead proposes a compositional diffusion planning approach that generalizes to much longer horizon via generative trajectory stitching.

**Trajectory Stitching.** A flurry of works have explored the trajectory stitching problem given offline data. One typical category of solution is based on data augmentation or goal relabeling along with various techniques, such as generative models [33, 41, 28, 30, 34, 40], model-based approaches [6, 32, 85, 23], and clustering [21]. Other supervised learning based methods, such as sequence modeling [9, 25, 73, 87, 71], latent space learning [80, 66, 83], and learning with dynamic programming [18, 27, 74], have also demonstrated some extent of stitching capability. In this work, we propose a different trajectory stitching approach based on generative modeling, where the model only learns from plain short trajectory segments while is able to directly perform goal-conditioned trajectory stitching through test-time compositional generation.

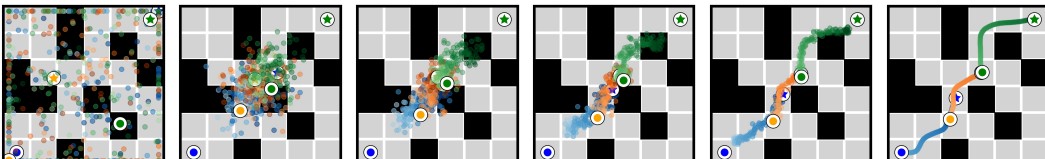

Figure 2: **Illustrating the Trajectory Stitching Process.** Given an unseen start (blue circle) and goal (green star), CompDiffuser generates a long-horizon plan by progressively denoising three trajectory chunks in parallel, with each chunk conditioning on its neighbors to ensure smooth transitions.

**Compositional Generative Models.** Compositional Generative Models [13, 11, 19, 12, 46, 3, 50, 63] are widely studied in various domains, including visual content generation [39, 11, 77, 82, 59, 57], human motion generation [56, 61, 81], traffic generation [38], robotic planning [78, 2, 43], and policy learning [69, 53]. Most existing work on compositionality focuses on sampling under the conjunction of several given conditions. While several studies [48, 47] have explored sequential compositional models, they are restricted to planning within a given skeleton and hence unable to generalize to longer sequences or new tasks. In contrast, we propose a compositional planning framework that scales to generating much longer sequences and completing new tasks via directly stitching short trajectory segments, without relying on pre-defined task-dependent skeletons.

## 3 Planning through Compositional Trajectory Generation

We aim to develop a generative planning framework that can generate long-horizon trajectories by composing multiple modular generative models. Our method, Compositional Diffuser (CompDiffuser), trains a single diffusion model on short-horizon trajectories. At inference time, given a start and goal, CompDiffuser runs parallel instances of this model to generate a sequence of overlapping trajectory segments, coordinating their denoising processes to ensure they smoothly connect into a coherent long-horizon plan. This approach enables us to stitch together short-horizon subsequences of training trajectories to form novel long-horizon trajectory plans.

### 3.1 Compositional Trajectory Modeling

Given a planning problem, consisting of a start state $q_s$ and a goal state $q_g$, we formulate planning as sampling a trajectory $\tau$ from the probability distribution

$$[s^{1:T}, a^{1:T}] \sim p_\theta(\tau | q_s, q_g), \tag{1}$$

where $s^{1:T}$ corresponds to future states to reach the goal state $q_g$ and $a^{1:T}$ corresponds to a set of future actions. To implement this sampling procedure, prior work [1, 26] learns a generative model $p(\tau)$ directly over previous trajectories $\tau$ in the environment. However, since the generative model is trained to model the density of previously seen trajectories, it is restricted to generating plans with start and goal that are similar to those seen in the past.

In this paper, we propose to model the generative model over trajectories $p_\theta(\tau | q_s, q_g)$ compositionally [12], where we subdivide trajectory $\tau$ into a set of $K$ overlapping sub-chunks $\tau_k$ (Figure 1). We then represent the trajectory distribution as

$$p_\theta(\tau | q_s, q_g) \propto p_1(\tau_1 | q_s, \tau_2) \, p_K(\tau_K | \tau_{K-1}, q_g) \prod_{k=2}^{K-1} p_k(\tau_k | \tau_{k-1}, \tau_{k+1}). \tag{2}$$

In the above expression, each trajectory chunk $\tau_k$ is only dependent on nearby trajectory chunks $\tau_{k-1}$ and $\tau_{k+1}$. This allows $p_\theta(\tau | q_s, q_g)$ to generate trajectory plans that significantly depart from previously seen trajectories, as long as intermediate trajectory chunks $\tau_k$ have been seen. Overall, the goal of our method is to enable long-horizon planning without long-horizon training data.

### 3.2 Training Compositional Trajectory Models

One approach to represent the composed probability distribution in Equation 2 is to directly learn separate generative models to represent each conditional probability distribution. However, sampling from the composed distribution is challenging, as each individual trajectory chunk depends on the

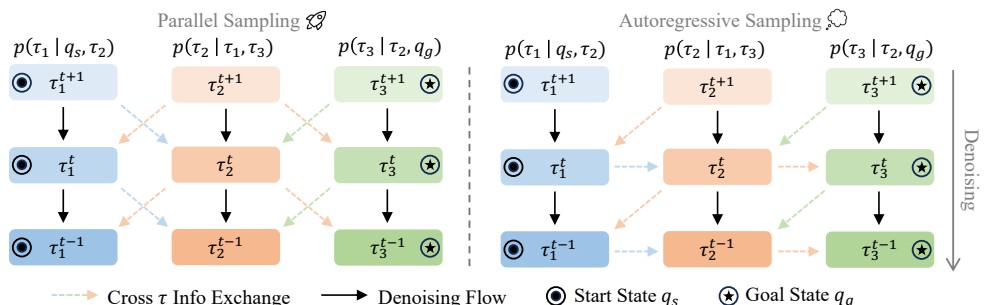

Figure 3: **Compositional Trajectory Planning: Parallel Sampling and Autoregressive Sampling**. We present an illustrative example of sampling three trajectories $\tau_{1:3}$ with the proposed compositional sampling methods. Dashed lines represent cross trajectory information exchange between adjacent trajectories and black lines represent the denoising flow of each trajectory. In parallel sampling, $\tau_{1:3}$ can be denoised concurrently; while in autoregressive sampling, denoising $\tau_k$ depends on the previous trajectory $\tau_{k-1}$, e.g., the denoising of $\tau_2$ depends on $\tau_1$ (as shown in the blue horizontal dashed arrows). Additionally, start state $q_s$ and goal state $q_g$ conditioning are applied to the trajectories in the two ends, $\tau_1$ and $\tau_3$, which enables goal-conditioned planning. Trajectories $\tau_{1:3}$ will be merged to form a longer plan $\tau_{\text{comp}}$ after the full diffusion denoising process.

values of neighboring chunks. As a result, to sample from the composed distribution, one would need a blocked Gibbs sampling procedure, where the value of each individual trajectory chunk is iteratively sampled given decoded values of neighboring trajectory chunks. This sampling procedure is slow, and passing information across chunks to form a consistent plan is challenging.

**Information Propagation Through Noisy-Sample Conditioning.** We propose a more efficient approach that addresses these challenges by leveraging the progressive denoising process of diffusion models. The key challenge in composing trajectories is ensuring feasible transitions between each pair of neighboring chunks, i.e., maintaining physical constraints and dynamic consistency at connection points where segments overlap. Our key insight is that we can achieve this by having trajectory segments guide each other's generation during the diffusion process: as one segment takes shape through denoising, it helps shape its neighbors into compatible configurations.

We implement this insight using a diffusion model that generates trajectory chunks conditioning on their neighbors' *noisy samples*. Given a dataset $D$ of trajectories $\tau$, we train a denoising network $\epsilon_\theta$ to learn the trajectory distribution $p_\theta(\tau_k | \tau_{k-1}, \tau_{k+1})$ with the training objective

$$\mathcal{L}_{\text{nbr}} = \mathbb{E}_{\tau \in D, t, k} \left[ \| \epsilon - \epsilon_\theta(\tau_k^t, t \mid \tau_{k-1}^t, \tau_{k+1}^t) \|^2 \right], \tag{3}$$

where $k$ identifies a trajectory segment, $t$ is the noise level, and $\tau_k^t$ represents segment $k$ corrupted with noise level $t$. Crucially, when denoising each segment, the network conditions on noisy versions of neighboring segments $\tau_{k-1}^t, \tau_{k+1}^t$ at the same noise level. This allows each segment to influence its neighbors' denoising process, ensuring their final configurations are dynamically compatible. In addition, further training the network to condition on $\tau_{k-1}^{t-1}$ can enable autoregressive compositional sampling, which we will discuss in Section 3.3. In practice, we only need to condition on the small overlapping regions between consecutive trajectories, making the generation process efficient while maintaining consistency across connection points.

**Representing Initial States and Goals.** In addition, we further train the same denoising network to represent the distributions $p_\theta(\tau_1 | q_s, \tau_2)$ and $p_\theta(\tau_K | \tau_{K-1}, q_g)$. This corresponds to the training objective

$$\mathcal{L}_{\text{start}} = \mathbb{E}_{\tau \in D, t, k} \left[ \| \epsilon - \epsilon_\theta(\tau_1^t, t \mid q_s, \tau_2^t) \|^2 \right], \tag{4}$$

with an analogous objective for the conditioned goal state $q_g$. We train the same denoising network $\epsilon_\theta$ with both conditioning. Please see Appendix E for implementation details. We provide an overview of our proposed training strategy in Algorithm 1.

### 3.3 Compositional Trajectory Planning

Our compositional framework enables flexible sampling strategies for generating long-horizon plans with Equation 2. The basic sampling process starts by initializing each trajectory chunk $\tau_k$ with Gaussian noise. Then, through iterative denoising, each chunk is denoised while being conditioned

**Algorithm 1** Training CompDiffuser

1: **Require:** train dataset $D$, diffusion denoiser model $\epsilon_\theta(\tau^t, t \mid \text{st\_cond}, \text{end\_cond})$, number of training steps $N$, diffusion timestep $T$
2: **for** $i = 1 \to N$ **do**
3: $\quad \tau^0 \sim D$, sample clean trajectory from dataset
4: $\quad \tau_{1:K}^0 \leftarrow$ divide $\tau^0$ to $K$ overlapping chunks
5: $\quad \tau_{1:K}^t \leftarrow$ add noise to $\tau_{1:K}^0$, $t \sim T$, following DDPM
6: $\quad$ # Objective for noisy chunk condition
7: $\quad \mathcal{L}_{\text{nbr}} = \|\epsilon - \epsilon_\theta(\tau_k^t, t \mid \tau_{k-1}^t, \tau_{k+1}^t)\|^2$, $k \sim [2, K-1]$
8: $\quad$ # Objective for start / end state condition
9: $\quad \mathcal{L}_{\text{start}} = \|\epsilon - \epsilon_\theta(\tau_1^t, t \mid q_s, \tau_2^t)\|^2$, $q_s = \tau_1^0[0]$
10: $\quad \mathcal{L}_{\text{end}} = \|\epsilon - \epsilon_\theta(\tau_K^t, t \mid \tau_{K-1}^t, q_g)\|^2$, $q_g = \tau_K^0[-1]$
11: $\quad \mathcal{L}_{\text{all}} = \mathcal{L}_{\text{nbr}} + \mathcal{L}_{\text{start}} + \mathcal{L}_{\text{end}}$
12: $\quad$ Backprop to update $\epsilon_\theta(.)$ using $\mathcal{L}_{\text{all}}$
13: **end for**
14: **return** $\epsilon_\theta(.)$

**Algorithm 2** Autoregressive Trajectory Sampling

1: **Models:** trained diffusion denoiser model $\epsilon_\theta(\tau, t \mid \text{st\_cond}, \text{g\_cond})$
2: **Input:** start state $q_s$, goal state $q_g$, number of composed trajectories $K$
3: Initialize $K$ trajectories $\tau_{1:K} \sim \mathcal{N}(0, I)$
4: **for** $t = T \to 1$ **do**
5: $\quad$ # Denoise $\tau_1$ conditioned on $q_s$ and $\tau_2^t$
6: $\quad \tau_1^{t-1} = \epsilon_\theta(\tau_1^t, t \mid q_s, \tau_2^t)$
7: $\quad$ # Denoise intermediate trajectories $\tau_2$ to $\tau_{K-1}$
8: $\quad$ **for** $k = 2 \to K-1$ **do**
9: $\quad\quad \tau_k^{t-1} = \epsilon_\theta(\tau_k^t, t \mid \tau_{k-1}^{t-1}, \tau_{k+1}^t)$
10: $\quad$ **end for**
11: $\quad$ # Denoise $\tau_K$ conditioned on $\tau_{K-1}^{t-1}$ and $q_g$
12: $\quad \tau_K^{t-1} = \epsilon_\theta(\tau_K^t, t \mid \tau_{K-1}^{t-1}, q_g)$
13: **end for**
14: $\tau_{\text{comp}} =$ Merge the denoised trajectories $\tau_{1:K}^0$
15: **return** $\tau_{\text{comp}}$

on its neighbors. This structure allows for different ways of coordinating the denoising process across trajectory chunks, each offering different tradeoffs between information propagation and computational efficiency. We present two sampling schemes (illustrated in Figure 3):

**Parallel Sampling**. Our first sampling approach conditions denoising on the values of the noisy adjacent trajectory chunks from the previous denoising timestep, where the update rule is

$$\tau_k^{t-1} = \alpha^t(\tau_k^t - \epsilon_\theta(\tau_k^t | \tau_{k-1}^t, \tau_{k+1}^t) + \beta^t \xi), \quad \xi \sim \mathcal{N}(0,1), \tag{5}$$

where $\alpha^t$ and $\beta^t$ are diffusion specific hyperparameters. This approach allows us to run denoising on each trajectory chunk in parallel, as each denoising update only requires the values of the adjacent trajectory chunks at a previous noise level. However, information propagation between the values of adjacent trajectory chunks is limited at each denoising timestep, as each trajectory chunk is denoised independently of the denoising updates of other trajectory chunks.

**Autoregressive Sampling**. To better couple the values of adjacent trajectory chunks, we propose to denoise each trajectory chunk autoregressively dependent on the values of neighboring chunks at each denoising timestep. In particular, we iteratively denoise each trajectory $\tau_{1:K}^t$ starting from the $\tau_1^t$, and condition the denoising of $\tau_k^t$ on the previously decoded chunk $\tau_{k-1}^{t-1}$ at the current noise level $t-1$ and the future chunk $\tau_{k+1}^t$ at the previous noise level $t$, giving us the sampling equation

$$\tau_k^{t-1} = \alpha^t(\tau_k^t - \epsilon_\theta(\tau_k^t | \tau_{k-1}^{t-1}, \tau_{k+1}^t) + \beta^t \xi), \quad \xi \sim \mathcal{N}(0,1). \tag{6}$$

This sequential generation process enables stronger coordination among the chunks since each chunk is conditioned on the less noisy version of its previous chunk. However, it requires generating chunks one at a time rather than simultaneously, making it computationally less efficient than parallel sampling. We compare the two sampling schemes in Table 6, where we empirically find autoregressive sampling leads to improved performance. Additionally, we provide sampling time comparison in Table 10. We use this autoregressive sampling procedure throughout the experiments in the paper and illustrate pseudocode for sampling in Algorithm 2. Given such final set of generated chunks $\tau_{1:K}$, we then merge the chunks together to construct a final trajectory $\tau_{\text{comp}}$ by applying exponential trajectory blending to areas where subchunks $\tau_k$ overlap (See Appendix E.2 for implementation details).

## 4 Experiments

In this section, our objective is to (1) validate that our method enforces coherent trajectory stitching on multiple benchmarks, with varying state space dimensions, task design, and training data collection policies (2) understand how planning with higher state dimensions, varying numbers of composed trajectories, different sampling schemes, and replanning affect the performance of the proposed method. Additional results are provided in Appendix B and C with failure analysis in Appendix D.

**Baselines.** We compare CompDiffuser with three categories of existing methods: (1) for generative planning methods, we include Decision Diffuser (DD) [1] for monolithic trajectory sampling

| Env | Size | RvS | RvS (SA) | RvS (GA) | DT | DT (SA) | DT (GA) | DD | GSC | Ours |
|---|---|---|---|---|---|---|---|---|---|---|
| PointMaze [21] | U-Maze | $17_{\pm7}$ | $97_{\pm5}$ | $76_{\pm5}$ | $17_{\pm5}$ | $65_{\pm4}$ | $54_{\pm4}$ | $0_{\pm0}$ | $\mathbf{100}_{\pm0}$ | $\mathbf{100}_{\pm0}$ |
| | Medium | $1_{\pm2}$ | $55_{\pm3}$ | $21_{\pm3}$ | $20_{\pm2}$ | $55_{\pm3}$ | $62_{\pm2}$ | $30_{\pm1}$ | $93_{\pm1}$ | $\mathbf{100}_{\pm0}$ |
| | Large | $3_{\pm4}$ | $38_{\pm5}$ | $31_{\pm5}$ | $22_{\pm2}$ | $35_{\pm2}$ | $39_{\pm5}$ | $0_{\pm0}$ | $99_{\pm2}$ | $\mathbf{100}_{\pm0}$ |

| Env | Type | Size | GCBC | GCIVL | GCIQL | QRL | CRL | HIQL | GSC | Ours |
|---|---|---|---|---|---|---|---|---|---|---|
| PointMaze [51] | stitch | Medium | $23_{\pm18}$ | $70_{\pm14}$ | $21_{\pm9}$ | $80_{\pm12}$ | $0_{\pm1}$ | $74_{\pm6}$ | $\mathbf{100}_{\pm0}$ | $\mathbf{100}_{\pm0}$ |
| | | Large | $7_{\pm5}$ | $12_{\pm6}$ | $31_{\pm2}$ | $84_{\pm15}$ | $0_{\pm0}$ | $13_{\pm6}$ | $\mathbf{100}_{\pm0}$ | $\mathbf{100}_{\pm0}$ |
| | | Giant | $0_{\pm0}$ | $0_{\pm0}$ | $0_{\pm0}$ | $50_{\pm8}$ | $0_{\pm0}$ | $0_{\pm0}$ | $29_{\pm3}$ | $\mathbf{68}_{\pm3}$ |

Table 1: **Quantitative Results on PointMaze Stitching Datasets in Ghugare et al. [21] and OGBench [51].** We compare CompDiffuser to baselines of multiple categories, including diffusion, data augmentation, and offline reinforcement learning. SA and GA stand for state augmentation and goal augmentation respectively, as described in Ghugare et al. [21]. Our results are averaged over 5 seeds and standard deviations are shown after the $\pm$ sign.

and Generative Skill Chaining (GSC) [48] for compositional sampling; (2) for data augmentation based methods, we include stitching specific data augmentation [21] with RvS [14] and Decision Transformer (DT) [9]; (3) for offline reinforcement learning methods, we include goal-conditioned behavioral cloning (GCBC) [44, 20], goal-conditioned implicit V-learning (GCIVL) and Q-learning (GCIQL) [29], Quasimetric RL (QRL) [70], Contrastive RL (CRL) [15], and Hierarchical implicit Q-learning (HIQL) [52]. See Appendix F for more details of baselines.

**Evaluation Setup.** For each environment, we report the success rate over all evaluation episodes, where the success criterion is that the agent or target object is close to the goal within a small threshold. We evaluate all methods with 5 random seeds for each experiment and report the mean and standard deviation. Specifically, in Ghugare et al. [21] datasets, we evaluate on 2 tasks in U-Maze, 6 tasks in Medium, and 7 tasks in Large with 10 episodes per task; in OGBench [51], we evaluate on 5 tasks in each environment with 20 episodes per task. Each task is introduced in the respective papers and is defined by a base start and goal state that require trajectory stitching to complete. A random noise is added to the base start and goal state for each evaluation episode.

## 4.1 PointMaze

We present experiment results on two types of trajectory-stitching datasets in point maze environments, which features different dataset collection strategies. Our method is trained on short trajectories of *x-y* positions of the point agent and is able to directly generate much longer trajectories from start and goal by composing multiple trajectories (detailed numbers of the composed trajectories in each experiment are provided in Appendix Table 13). Note that our method only requires training one model and we can use the proposed compositional inference method to construct coherent long-horizon trajectories.

**PointMaze** [21]. Following the original framework, the training data are curated by dividing each environment (here, maze) into several small regions, and feasible trajectories constrained within their respective regions are sampled. Possible stitching between segments is facilitated by a small overlap (one block) between different regions, which can be used to join trajectories across regions. More dataset details are provided in Appendix A.1. We present the quantitative results on these datasets in Table 1, where we compare to goal-conditioned behavior cloning methods trained with data augmentation, Decision Diffuser, and GSC. Most baselines perform suboptimally, likely because they are unable to autonomously identify the small overlapping regions needed for trajectory stitching. In contrast, CompDiffuser successfully resolves all tasks across various maze sizes, demonstrating its ability to stitch trajectories even when the connecting regions are small.

**PointMaze in OGBench** [51]. In this particular setup, each trajectory in the training dataset is constrained to navigate no more than four blocks in the environment. The start and goal of each trajectory can be sampled from the entire environment provided that the travel distance between the start and goal is within four blocks. These trajectories are much shorter than the ones required for a feasible plan between a given start and goal at inference. More details about these datasets are provided in Appendix A.2. We present the quantitative results in Table 1, where we compared our method to GSC and multiple offline RL baselines following the benchmark established in [51]. We observe that while GSC is able to perform on par with CompDiffuser in Medium and Large mazes, it struggles as the planning horizon further increases, as illustrated in the qualitative results in Figure 4.

| Env | Type | Size | GCBC | GCIVL | GCIQL | QRL | CRL | HIQL | GSC | Ours |
|---|---|---|---|---|---|---|---|---|---|---|
| **antmaze** | stitch | Medium | 45 ±11 | 44 ±6 | 29 ±6 | 59 ±7 | 53 ±6 | **94** ±1 | **97**±2 | **96**±2 |
| | | Large | 3 ±3 | 18 ±2 | 7 ±2 | 18 ±2 | 11 ±2 | 67 ±5 | 66±2 | **86**±2 |
| | | Giant | 0 ±0 | 0 ±0 | 0 ±0 | 0 ±0 | 0 ±0 | 21 ±2 | 20±1 | **65**±3 |
| | explore | Medium | 2 ±1 | 19 ±3 | 13 ±2 | 1 ±1 | 3 ±2 | 37 ±10 | **90**±2 | 81±2 |
| | | Large | 0 ±0 | 10 ±3 | 0 ±0 | 0 ±0 | 0 ±0 | 4 ±5 | 21±3 | **27**±1 |
| **humanoid maze** | stitch | Medium | 29 ±5 | 12 ±2 | 12 ±3 | 18 ±2 | 71 ±3 | **96** ±4 | 92±1 | 91±1 |
| | | Large | 6 ±3 | 1 ±1 | 0 ±0 | 3 ±1 | 6 ±1 | 31 ±3 | 70±3 | **72**±3 |
| | | Giant | 0 ±0 | 0 ±0 | 0 ±0 | 0 ±0 | 0 ±0 | 12 ±2 | 5±1 | **67**±4 |

Table 2: **Quantitative Results on AntMaze and HumanoidMaze in OGBench.** We benchmark our method on the 5 test-time tasks defined in OGBench with 20 episodes per task. Our results are averaged over 5 seeds and standard deviations are shown after the ± sign.

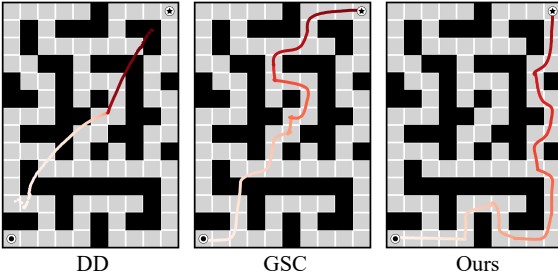

DD    GSC    Ours    Top-down Plan Overview    Zoom-in

Figure 4: **Qualitative Comparison of DD, GSC and Comp-Diffuser on OGBench PointMaze Giant.** Effective bidirection information propagation enables CompDiffuser to successfully synthesize trajectories from start (bottom left) to goal (upper right), while other methods generate *o.o.d* trajectories that are disconnected with the start/goal or passing through walls. See Figure 10 for per-segment visualization.

Figure 5: **Qualitative Results of Planning in High Dimension on OGBench AntMaze Large.** Original plan is sub-sampled for clearer view. Our method is able to synthesize plans of valid dynamics while reaching the goal position (bottom right). Note that our method is only trained on trajectory segments of much shorter length.

CompDiffuser significantly outperforms all baselines in Giant maze, demonstrating its efficacy in complex environments. Additional results are provided in Appendix C.1, C.2, and C.3.

## 4.2 High Dimension Tasks

We present the evaluation results of our method on various trajectory stitching tasks involving higher-dimensional state spaces within OGBench: AntMaze, HumanoidMaze, and AntSoccer.

**AntMaze and HumanoidMaze.** We conduct maze navigation experiments of multiple agents, ant and humanoid, using the pre-collected Stitch datasets provided in OGBench. The data collection strategy is identical to OGBench PointMaze, where each episode is constrained to travel at most 4 blocks, while at inference, a successful plan requires the agent to travel up to 30 blocks. We compare our method with the offline RL benchmark established in [51] along with the best-performing diffusion-based compositional stitching baseline GSC, as shown in Table 2. Both GSC and CompDiffuser generate plans in a planar *x-y* space while the agent follows the plans with a learned inverse dynamics model. We observe a pattern similar to the results in PointMaze, where CompDiffuser can consistently give high success rates as the planning horizon and complexity increase while other baselines start to collapse. To complete the study, we also conduct experiments where the full agent state is used for planning instead of the *x-y* space and provide the results in Section 4.3.

**AntMaze with Low-Quality Data.** We evaluate CompDiffuser on OGBench AntMaze with a different data collection strategy, Explore. These datasets consist of extremely low-quality yet high-coverage data, where the data collection policy contains a large amount of action noise and will randomly re-sample a new moving direction after every 10 steps (see Figure 9 in Appendix for qualitative examples). Hence, each demonstration episode typically oscillates within only 2-3 blocks. Our planner needs to learn from these clustered trajectories to construct coherent plans that reach goals in large spatial distances. We present the success rate of each method in Table 2.

**AntSoccer.** The AntSoccer environment in OGBench requires the ant agent to move a soccer ball to a designated goal in the environment, different from maze tasks that require the agent itself to

| Env | Size | GCBC | GCIVL | GCIQL | QRL | CRL | HIQL | GSC (4D) | Ours (4D) | GSC (17D) | Ours (17D) |
|---|---|---|---|---|---|---|---|---|---|---|---|
| antsoccer | arena | $34_{\pm4}$ | $21_{\pm3}$ | $5_{\pm2}$ | $2_{\pm1}$ | $2_{\pm1}$ | $23_{\pm2}$ | $41_{\pm4}$ | $55_{\pm6}$ | $65_{\pm3}$ | $\mathbf{69}_{\pm3}$ |
| stitch | medium | $2_{\pm1}$ | $1_{\pm0}$ | $0_{\pm0}$ | $0_{\pm0}$ | $0_{\pm0}$ | $8_{\pm2}$ | $5_{\pm2}$ | $13_{\pm1}$ | $12_{\pm2}$ | $\mathbf{17}_{\pm3}$ |

Table 3: **Quantitative Results on OGBench AntSoccer Stitch.** We evaluate two generative planners with different planning state dimensions: a 4D planner that operates on the *x-y* positions of the ant and the ball, and a 17D planner that additionally generates the 13 joint positions of the ant agent.

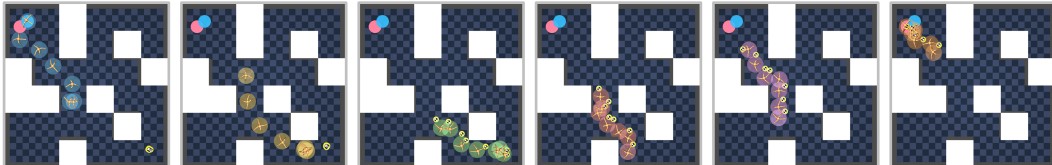

Figure 6: **Qualitative Plan generated by CompDiffuser in OGBench AntSoccer Medium Stitch.** The initial position of the ant is shown by the blue circle and the goal is to move the ball to the pink circle. We plot each individual trajectory $\tau_{1:6}$ separately (as shown from left to right) and mark the ball with a yellow circle for clearer view. These generated trajectories will be merged to form a long-horizon plan $\tau_{\text{comp}}$. Note that our model is only trained on two different types of short trajectories: 1) the ant moves without dribbling the ball; 2) the ant moves while dribbling the ball. At test time, the proposed compositional sampling method can stitch these two types of trajectories and construct end-to-end plans of longer horizon to solve the more difficult task – first navigate to the ball from the far side and then dribble the ball to the goal position.

reach the goal. OGBench provides two categories of trajectories in `AntSoccer Stitch` datasets: (1) ant navigates in the maze without the ball and (2) ant moves and dribbles the ball in the maze. The planner must stitch trajectories of these two skills to complete the goal-reaching objective, because the ant must reach the ball first and then dribble it to the given goal location (see Figure 6 and Figure 14).

We present experimental results in two distinct maze configurations, `Arena` and `Medium`, in Table 3 relative to the benchmarking provided in OGBench. Specifically, as an ablation study, we evaluate with two planner state space configurations: (1) a 4D planner consisting of the *x-y* location of the ant and the ball; (2) a 17D planner consisting of the *x-y* location of the ant and the ball along with all the joint positions of the ant. We observe that our compositional method outperforms all the baselines in both configurations. Notably, the 17D variant demonstrates slightly higher success rates, likely due to the ability of joint positions to provide more fine-grained information for ball dribbling.

## 4.3 Ablation Studies

**Planning in High Dimension Space.** We report experiment results where CompDiffuser synthesizes trajectories in state space of higher dimension. We compare the success rates of our method planning in dimension of 2D, 15D, and 29D in Table 4, and present qualitative plans in Figure 5 and Figure 13. Specifically, the planner operates in the *x-y* space for 2D, *x-y* with the joint positions for 15D, *x-y* with both the joint positions and velocities for 29D. Similar to Section 4.2, we train an MLP inverse dynamics model that takes in the current observation and a goal of the planner dimension to predict an action for the agent to execute.

Our method performs consistently and achieves near-optimal success rates across all planning dimensions in `AntMaze Medium`. In `AntMaze Large` and `Giant`, the success rates decrease when planning in 15D and 29D, which is probably due to the increasing complexity of the trajectory modeling, since the joint positions and velocities of a moving ant are highly dynamic and the tasks require planning hundreds of consecutive future states to reach the goal.

**Different Numbers of Composed Trajectories.** We study the effect of varying the numbers of trajectories to be composed $K$. To better study the planning performance with respect to $K$, we use the challenging `PointMaze-Giant-Stitch` in OGBench as the testbed. As shown in Figure 7, our method obtains consistent performance when composing 7 to 12 trajectories. We observe that the optimal $K$ is around 9 and 10, which also corresponds to a natural path length to reach the goal. Qualitatively, decreasing $K$ will result in a sparser trajectory while increasing $K$ will cause the final trajectory traveling back and forth to consume the redundant states (See Figure 10).

| Size | HIQL | Ours (2D) | Ours (15D) | Ours (29D) |
|---|---|---|---|---|
| Medium | $94_{\pm 1}$ | $96_{\pm 2}$ | $95_{\pm 0}$ | $\mathbf{97}_{\pm 2}$ |
| Large | $67_{\pm 5}$ | $\mathbf{86}_{\pm 2}$ | $66_{\pm 5}$ | $66_{\pm 5}$ |
| Giant | $21_{\pm 2}$ | $\mathbf{65}_{\pm 3}$ | $41_{\pm 3}$ | $28_{\pm 4}$ |

Table 4: **Quantitative Results of Different Planning Dimensions on AntMaze Stitch Datasets in OGBench.** Our method compositionally constructs feasible plans that reach long-distance goals while modeling complex environment dynamics, such as the positions and velocities of the agent's joints.

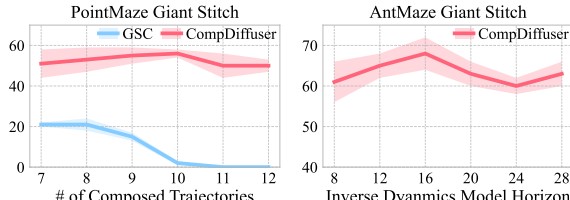

Figure 7: **Quantitative Results of Different Setup.** Left: success rate versus different numbers of composed trajectories $K$ in OGBench `PointMaze Giant` (w/o replan). Right: success rate versus different inverse dynamics model horizons in OGBench `AntMaze Giant`.

| Env | Size | QRL | HIQL | Ours w/o Replan | Ours w/ Replan |
|---|---|---|---|---|---|
| PointMaze | Medium | $80_{\pm 12}$ | $74_{\pm 6}$ | $\mathbf{100}_{\pm 0}$ | $\mathbf{100}_{\pm 0}$ |
| | Large | $84_{\pm 15}$ | $13_{\pm 6}$ | $\mathbf{100}_{\pm 0}$ | $\mathbf{100}_{\pm 0}$ |
| | Giant | $50_{\pm 8}$ | $0_{\pm 0}$ | $\mathbf{53}_{\pm 6}$ | $68_{\pm 3}$ |
| AntMaze | Medium | $59_{\pm 7}$ | $\mathbf{94}_{\pm 1}$ | $92_{\pm 2}$ | $96_{\pm 2}$ |
| | Large | $18_{\pm 2}$ | $67_{\pm 5}$ | $\mathbf{76}_{\pm 2}$ | $86_{\pm 2}$ |
| | Giant | $0_{\pm 0}$ | $21_{\pm 2}$ | $\mathbf{27}_{\pm 4}$ | $65_{\pm 3}$ |

Table 5: **Quantitative Results of CompDiffuser with and without Replanning.** We report the success rates on OGBench `PointMaze` and `AntMaze Stitch` datasets. For w/o replan, CompDiffuser only synthesizes one trajectory and executes the trajectory in a close-loop manner; for w/ replan, CompDiffuser will synthesize a new trajectory if the agent loses track of the current plan.

| Env | Size | Replan | Parallel | AR |
|---|---|---|---|---|
| PointMaze | Large | ✓ | $\mathbf{100}_{\pm 0}$ | $\mathbf{100}_{\pm 0}$ |
| | Giant | ✗ | $45_{\pm 1}$ | $53_{\pm 6}$ |
| | Giant | ✓ | $66_{\pm 2}$ | $68_{\pm 3}$ |
| AntMaze | Large | ✓ | $84_{\pm 5}$ | $86_{\pm 2}$ |
| | Giant | ✗ | $18_{\pm 4}$ | $27_{\pm 4}$ |
| | Giant | ✓ | $48_{\pm 1}$ | $65_{\pm 3}$ |

Table 6: **Quantitative Comparison of two Sampling Schemes: Parallel and Autoregressive (AR).** Parallel sampling performs on par with AR sampling in the easier `Large` maze, while AR sampling can generate trajectories of higher quality when constructing longer plans (i.e., composing more trajectories), likely due to its causal denoising strategy.

**Replanning with CompDiffuser.** Our method can also flexibly replan during a rollout, which enables the agent to recover from failure, such as when the agent fails to track the planned trajectory due to sub-optimal inverse dynamic actions. In practice, we replan if the distance between the agent's current observation and the synthesized subgoal is larger than a threshold. Please see Appendix E.3 for details. In Table 5, we present ablation studies of CompDiffuser with and without replanning in `PointMaze` and `AntMaze Stitch` datasets in OGBench. CompDiffuser outperforms the best performing baselines QRL and HIQL even without replanning in 5 out of 6 tasks. We also observe that w/ and w/o replanning yield similar performance in maze size `Medium` and `Large`, while offering significant performance boost in the more complex `Giant` maze.

**Parallel vs. Autoregressive Sampling.** We compare the performance of the two proposed compositional sampling schemes, parallel and autoregressive, as presented in Table 6. Autoregressive sampling consistently outperforms the parallel sampling across various tasks in plan quality, showing that the causal information flow, where each trajectory chunk is conditioned on the already-denoised (less noisy) version of previous chunk, leads to more coherent and physically consistent transitions between chunks. We include additional discussion and sampling time comparison in Appendix B.3.

## 5 Discussion and Conclusion

**Limitations.** Stitching short trajectories to solve for unseen longer-horizon plans is a challenging problem. While our method excels in our empirical evaluation, there still remain a number of future venues of research. When composing a large sequence of trajectories, the error accumulation in the long chain of bidirectional information propagation might lead to infeasible plans. This issue can be potentially mitigated by using domain/task specific rejection sampling techniques to select from multiple candidate plans or leveraging more expensive MCMC sampling. Moreover, the optimal number of the test-time composed chunks $K$ is task-dependent. Our future research will explore ways to automatically identify a suitable number of chunks $K$, potentially by incrementally increasing the number of chunks dependent on the quality of the generated plan.

**Conclusion.** We introduce CompDiffuser, a generative trajectory stitching method that leverages the compositionality of diffusion models. We introduce a noise-conditioned score function formulation that helps in performing autoregressive sampling of multiple short-horizon trajectory diffusion models and eventually stitching them to form a longer-horizon goal-conditioned trajectory. Our method demonstrates effective trajectory stitching capabilities as evident from the extensive experiments on tasks of various difficulty, including different environment sizes, planning state dimensions, trajectory types, and training data quality.

## Acknowledgments

We would like to thank Zilai Zeng for helpful early discussion on trajectory stitching and feedback of the manuscript.

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

# Contents

# Appendix

In this appendix, we first introduce the evaluation environments and stitching datasets in Section A. Next, we provide additional quantitative results in Appendix B and additional qualitative results in Appendix C. We provide failure mode analysis in Section D. Following this, we provide implementation details in Section E, including model architecture, training and evaluation setups, trajectory merging, and replanning. Lastly, we introduce notable baselines in Section F.

## A   Environment and Stitching Datasets

In this paper, we directly evaluate our method on public stitching datasets introduced in two recent papers Ghugare et al. [21] and OGBench [51]. In this section, we provide detailed descriptions of each dataset along with qualitative examples of trajectories in these datasets.

### A.1   Stitching Datasets in Ghugare et al. [21]

This paper [21] divides each evaluation environment into several small regions and each demonstration trajectory in the training datasets can only navigate within a specific region. There is a small overlap (one block) between each region, which can be used to stitch trajectories across regions. Therefore, to complete test-time goals, the agent needs to conduct effective reasoning based on the given start state and goal state and identify the corresponding overlap joints. The division of regions is visualized in the original paper [21]. We use the environments and datasets from their official implementation release at https://github.com/RajGhugare19/stitching-is-combinatorial-generalisation.

### A.2   OGBench Datasets

OGBench is a comprehensive benchmark designed for offline goal-conditioned RL. Since our focus is to evaluate the trajectory stitching ability of CompDiffuser, we use the `Stitch` and `Explore` dataset types in OGBench.

In `Stitch` datasets, trajectories are constrained to navigate no more than 4 blocks in the environment. The start and goal state of each trajectory can be sampled from the entire environment provided that the travel distance between the start and goal is within 4 blocks. Qualitative examples of the trajectories in the `Stitch` dataset are shown in Figure 8.

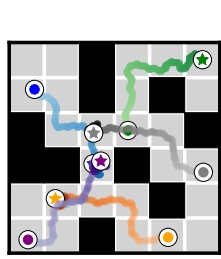 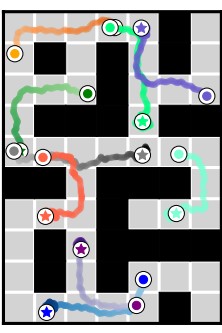 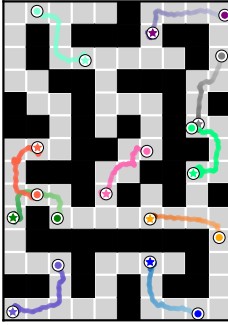

|  AntMaze Medium Stitch  |  AntMaze Large Stitch  |  AntMaze Giant Stitch  |

Figure 8: **Trajectory Examples in OGBench AntMaze Stitch Datasets.** Each Trajectory is limited to travel at most 4 blocks for dataset type `Stitch`, while at inference, the distance between the start and goal can be up to 30 in the `Giant` Maze.

In `Explore` datasets, trajectories are of extremely low-quality though high-coverage. The data collection policy contains a large amount of action noise and will randomly re-sample a new moving direction after every 10 steps. Hence, each demonstration trajectory in the training dataset typically moves within only 2-3 blocks due to the random moving direction. These datasets might be even more challenging due to the large noisy and cluster-like trajectory pattern. Qualitative examples of the trajectories in the `Explore` dataset are shown in Figure 9.

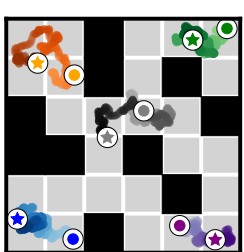 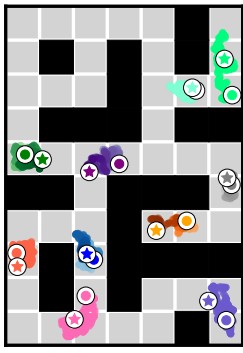

AntMaze Medium Explore      AntMaze Large Explore

Figure 9: **Trajectory Examples in OGBench AntMaze Explore Datasets.** Trajectories in `Explore` datasets are of extremely low-quality though high-coverage. The data collection policy contains a large amount of action noise and will randomly re-sample a new moving direction after every 10 steps.

We show the environment names, corresponding datasets, and the maximum environment steps for each evaluation episode in Table 7. All methods are trained on the OGBench public release datasets. We slightly increase the environment steps for some environments due to the task difficulty (e.g., `Giant` Maze). For these environments, we follow the implementation in https://github.com/seohongpark/ogbench. and rerun all baselines with the increased maximum environment steps; for other environments, we directly adopt the reported success rates in the original paper.

| Environment | Type | Size | Dataset Name | Env Steps |
|---|---|---|---|---|
| `pointmaze` | `stitch` | `Medium` | pointmaze-medium-stitch-v0 | 1000 |
| | | `Large` | pointmaze-large-stitch-v0 | 1000 |
| | | `Giant` | pointmaze-giant-stitch-v0 | 1000 |
| `AntMaze` | `Stitch` | `Medium` | antmaze-medium-stitch-v0 | 1000 |
| | | `Large` | antmaze-large-stitch-v0 | 2000 |
| | | `Giant` | antmaze-giant-stitch-v0 | 2000 |
| `AntSoccer` | `stitch` | `Arena` | antsoccer-arena-stitch-v0 | 5000 |
| | | `Medium` | antsoccer-medium-stitch-v0 | 5000 |
| `HumanoidMaze` | `Stitch` | `Medium` | humanoid-medium-stitch-v0 | 5000 |
| | | `Large` | humanoid-large-stitch-v0 | 5000 |
| | | `Giant` | humanoid-giant-stitch-v0 | 8000 |

Table 7: **Our Evaluation Environments and Datasets in OGBench.**

# B Additional Quantitative Results

In this section, we provide additional quantitative experiment results. In Section B.1, we study how varying the number of composed trajectories $K$ affects the performance of our method. In Section B.2, we investigate the effect of inverse dynamic models of different horizons. Next, we provide a sampling time comparison of Decision Diffuser and the proposed parallel and autoregressive sampling schemes in Section B.3. Following this, in Section B.4, we analyze how the proposed autoregressive sampling scheme performs when using different denoising starting directions.

## B.1 Number of Composed Trajectories

In Table 8, we compare CompDiffuser with GSC over composing different numbers of trajectories (results are also shown in Figure 7). Our method performs steadily when composing different numbers of trajectories while GSC collapses.

| | PointMaze Giant Stitch | | | | | |
|---|---|---|---|---|---|---|
| # Comp | 7 | 8 | 9 | 10 | 11 | 12 |
| GSC | $21_{\pm1}$ | $21_{\pm3}$ | $15_{\pm2}$ | $2_{\pm1}$ | $0_{\pm0}$ | $0_{\pm0}$ |
| CompDiffuser | $51_{\pm7}$ | $53_{\pm6}$ | $55_{\pm4}$ | $56_{\pm2}$ | $50_{\pm6}$ | $50_{\pm3}$ |

Table 8: **Quantitative Results over Different Numbers of Composed Trajectories.** We report success rates (w/o replanning) of composing 7 to 12 trajectories in the OGBench `PointMaze Giant Stitch` dataset. CompDiffuser can consistently construct feasible trajectories over various numbers of composed trajectories while GSC gradually collapses.

## B.2 Inverse Dynamics Model

In this section, we evaluate our planner with inverse dynamics models of different horizons (results are also shown in Figure 7). Specifically, we use an MLP to implement the inverse dynamics model, which takes as input the start and goal state and outputs an action. We use the same training dataset (as to train the planner) to train the corresponding inverse dynamics model. As shown in Table 9, our method performs steadily across different inverse dynamics model horizons, showing that the subgoals generated by our planners are of high feasibility and are robust to various inverse dynamics models' configurations.

| Env | Type | Size | 8 | 12 | 16 | 20 | 24 | 28 |
|---|---|---|---|---|---|---|---|---|
| AntMaze | Stitch | Large | $77_{\pm4}$ | $\mathbf{86}_{\pm2}$ | $80_{\pm2}$ | $85_{\pm3}$ | $76_{\pm2}$ | $77_{\pm3}$ |
| | | Giant | $61_{\pm5}$ | $65_{\pm3}$ | $\mathbf{68}_{\pm4}$ | $63_{\pm3}$ | $60_{\pm2}$ | $63_{\pm3}$ |

Table 9: **Quantitative Results of CompDiffuser with Inverse Dynamics Models of Different Horizons.** We present the success rates of CompDiffuser with 6 different inverse dynamics model horizons. In both environments, `Large` and `Giant`, our method performs consistently across all configurations, showing that the synthesized plans adhere to the transition dynamics and are easy to follow.

## B.3 Sampling Time Comparison: Parallel vs. Autoregressive

In Table 10, we compare the diffusion denoising sampling time of three methods: (1) monolithic model method, Decision Diffuser (DD), where we directly sample a long trajectory with the same horizon as the proposed compositional sampling method; (2) parallel compositional sampling, as shown in the left of Figure 3, where we denoise all trajectories $\tau_{1:K}$ in one batch; (3) autoregressive compositional sampling, as shown in the right of Figure 3, where we sequentially denoise each $\tau_k$.

We observe that both parallel and AR sampling methods require more sampling time than DD probably due to (1) the the simpler and smaller denoiser network of DD; (2) time for condition

encoding (our method will first encode the noisy adjacent trajectories $\tau_{k-1}$ and $\tau_{k+1}$ and feed the resulting latents to the denoiser $\epsilon_\theta$) and (3) the overhead of trajectory merging.

In addition, in our parallel sampling scheme, we stack all trajectories $\tau_{1:K}$ to one batch and feed it to the denoiser network $\epsilon_\theta$. While it indeed requires only one model forward, the batch size increases implicitly, which is probably the major reason that the sampling time of Ours (Parallel) does not proportionally decrease as the number of composed trajectories $K$, in comparison to Ours (AR).

| Env | Type | Size | DD (Monolithic) | Ours (Parallel) | Ours (AR) |
|---|---|---|---|---|---|
| PointMaze | Stitch | Medium | 0.23 | 1.02 | 1.54 |
| | | Large | 0.23 | 1.67 | 3.39 |
| | | Giant | 0.23 | 2.73 | 5.00 |

Table 10: **Quantitative Comparison of Sampling Time** We report the time for sampling one (compositional) trajectory in three different `PointMaze` environments using one Nvidia L40S GPU (unit: second). The reported results are averaged over 20 sampling. In Parallel and AR (Autoregressive) mode, we use the proposed compositional sampling scheme as shown in Figure 3. Specifically, we compose 3, 6, and 9 trajectories in Maze `Medium`, `Large`, and `Giant`, respectively. In Decision Diffuser (DD), we directly sample one trajectory with identical length as the compositional counterparts.

## B.4 Starting Direction of Autoregressive Compositional Sampling

In the proposed autoregressive sampling scheme described in Figure 3, inside each diffusion denoising timestep, the denoising starts from $\tau_1$ and sequentially proceeds to $\tau_K$ (from left to right), which we denote as forward passing. Another implementation variant is to denoise in the reverse order, that is, first denoise $\tau_K$ and sequentially proceed to $\tau_1$ (from right to left), which we denote as backward passing.

We provide a quantitative comparison of these two starting directions in Table 11. We observe that these two methods yield similar performance, demonstrating that either autoregressive sampling direction can enable effective information propagation and exchange.

| Env | Type | Size | Ours (Forward) | Ours (Backward) |
|---|---|---|---|---|
| PointMaze | Stitch | Medium | $100_{\pm 0}$ | $100_{\pm 0}$ |
| | | Large | $100_{\pm 0}$ | $100_{\pm 0}$ |
| | | Giant | $55_{\pm 4}$ | $56_{\pm 2}$ |

Table 11: **Quantitative Comparison of Forward and Backward Information Propagation.** We study the effect of the starting direction of the autoregressive sampling, from $\tau_1$ vs. from $\tau_K$. To directly study the trajectory quality, we report the results w/o replanning for both methods. Either Forward or Backward achieves similar performance, suggesting that our sampling method is robust to different sampling configurations.

# C    Additional Qualitative Results

In this section, we present additional qualitative results of CompDiffuser. Videos and interactive demos are provided at our project website (see *Abstract* section for the link).

## C.1    Composing Different Numbers of Trajectories

We present qualitative results of composing 8 to 11 trajectories in OGBench `PointMaze Giant Stitch` environments in Figure 10. We compositionally sample multiple trajectories to construct a long-horizon plan where the given start is at the bottom-left corner and the given goal is at the top-right corner. For clearer view, we present the results before applying trajectory merging (i.e., we show each individual trajectory of $\tau_{1:K}$) and use different colors to highlight different trajectories.

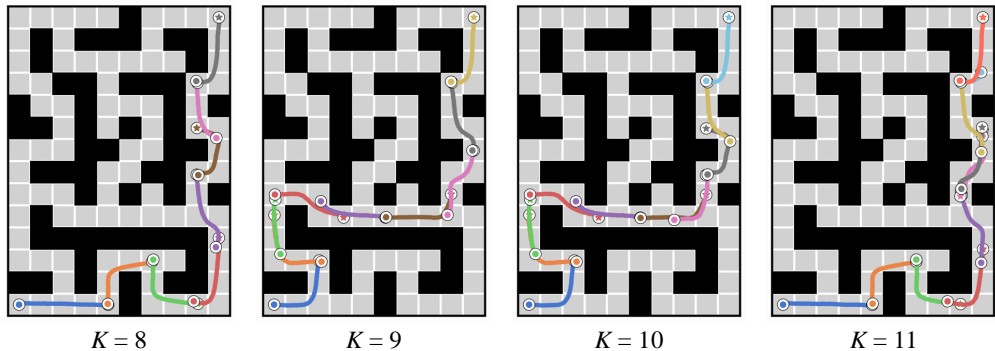

| $K = 8$ | $K = 9$ | $K = 10$ | $K = 11$ |

Figure 10: **Composing Different Numbers of Trajectories at Inference.** Our method is only trained on short trajectory segments that travel at most 4 blocks, while is able to compositionally generate coherent long trajectories given the start (circle) and goal (star). When $K$ is smaller and barely sufficient to reach the goal (e.g., 8), the length of the overlapping part between segments decreases so as to extend the travel distance of the overall compositional plan. In contrast, if given a larger $K$ (e.g., 11), some parts of the compositional plan might travel back and forth to consume the extra length.

## C.2    Diverse Trajectory Morphology

The proposed compositional sampling method allows direct generalization to long-horizon planning tasks at test time through its noisy-sample conditioning and bidirectional information propagation design. Meanwhile, this sampling approach also preserves the multi-modal nature of the diffusion model, enabling a diverse range of trajectory morphology. As shown in Figure 11, given a similar start and goal pair, our method can construct trajectories that reach the goal via various possible paths. With such multi-modal flexibility, the proposed sampling process can be further customized with specific preferences by integrating additional test-time steering techniques.

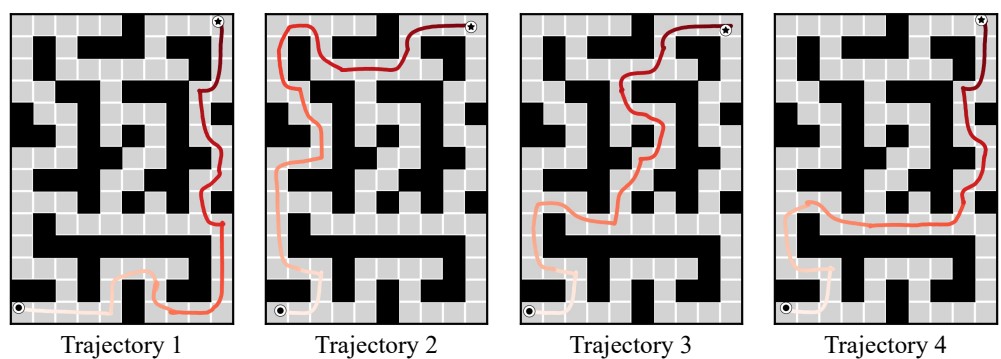

| Trajectory 1 | Trajectory 2 | Trajectory 3 | Trajectory 4 |

Figure 11: **Diverse Trajectory Morphology.** We present four trajectories with similar start and goal in OGBench `PointMaze Giant Stitch`. All trajectories are generated by CompDiffuser, composing 9 trajectories. CompDiffuser preserves the multi-modal nature of diffusion models and is able to flexibly sample trajectories of diverse morphology.

## C.3 Planning Results on Different Tasks.

In Section C.2 above, we present multiple trajectories of Task 1 in OGBench `PointMaze Giant`. In this section, we present qualitative results of the following Task 2 to Task 5, as in Figure 12. We set the number of composed trajectories $K$ to 9 in all tasks. The state state is shown by the black circle and the goal state is shown by the black star. Our method successfully constructs feasible plans for various start-goal configurations across different spatial distances.

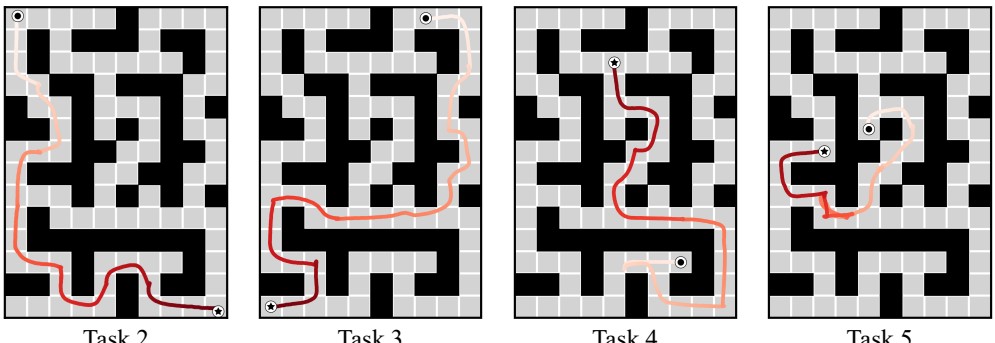

| Task 2 | Task 3 | Task 4 | Task 5 |

Figure 12: **Different Tasks in OGBench PointMaze Giant Stitch.** We present qualitative plans by CompDiffuser on OGBench Task 2-5 (See Figure 11 for results of Task 1). We set the number of composed trajectories to 9 for all the tasks above. The black circle denotes the start state and the black star denotes the goal state. In task 2 and 3, our method effectively constructs long horizon plans that reach the goals on the opposite side of the environment (despite being trained only on trajectories that travel at most 4 blocks). In comparison, Task 4 and 5 feature a relatively smaller spatial gap between the start and goal, thus requiring a shorter planning horizon. Our method still generalizes to these tasks by generating plans that traverse additional distance or leveraging back-and-forth movements to consume the extra plan length.

## C.4 Compositional Planning in High Dimensional Space

In this section, we present additional high dimensional trajectories generated by CompDiffuser in OGBench `AntMaze Large Stitch` environment. Similar to other experiments in the paper, the models are trained on the corresponding OGBench public release datasets.

**AntMaze Large Stitch 29D.** We train CompDiffuser on the state space of *x-y* position along with the ant's joint positions and velocities, resulting in a 29D planning task. Note that CompDiffuser is only trained on short trajectory segments (we set its horizon to 160), while at test time, we compose 5 trajectories to directly construct trajectory plans of horizon 584. We present additional qualitative results in Figure 13. Note that we uniformly sub-sample the length of the trajectory to 50 for clearer view. Corresponding quantitative results are reported in Table 4.

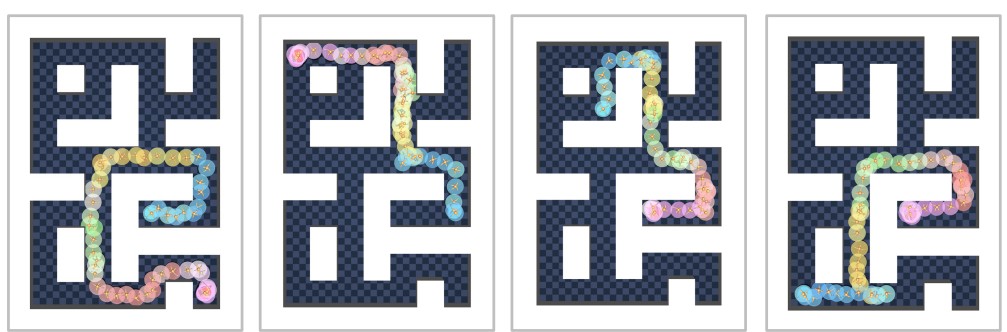

Figure 13: **Compositional Planning in 29D on OGBench AntMaze Large Stitch Tasks.** Original trajectory plans are much denser and we uniformly sub-sample 50 states from the original generated trajectories for better view. Five trajectories are composed as shown by different colors: the blue one indicates the first trajectory $\tau_1$ and the purple one indicates the fifth trajectory $\tau_5$.

## C.5 Compositional Planning in AntSoccer Arena

We provide additional qualitative results in OGBench `AntSoccer Arena Stitch` environment in Figure 14. In this task, the ant is initialized to the location of the blue circle and is tasked to move the ball to the goal location indicated by the pink circle. Hence, the ant needs to first reach the ball from the far side of the environment and dribble the ball to the goal.

However, such long-horizon trajectories (the ant reaches the ball and then dribbles the ball to the goal) do not exist in the training dataset. The training dataset only contains two distinct types of trajectories: (1) the ant moves in the environment without the ball, (2) the ant moves while dribbling the ball. Therefore, the planner needs to generalize and stitch in a zero-shot manner – constructing an end-to-end trajectory that first approaches the ball and then dribbles the ball to the goal.

We train CompDiffuser on the state space of the ant's $x$-$y$ position and joint positions along with the $x$-$y$ position of the ball, resulting in a 17D planning task. Similar to Figure 6, we present each individual trajectory $\tau_{1:K}$ in Figure 14 for clearer view. Corresponding quantitative results are provided in Table 3.

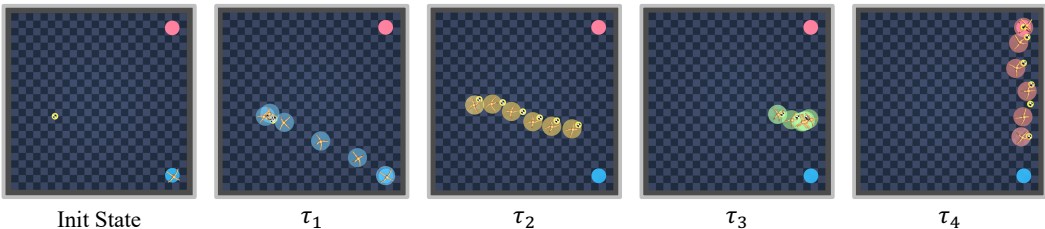

| Init State | $\tau_1$ | $\tau_2$ | $\tau_3$ | $\tau_4$ |

Figure 14: **Compositional Planning on OGBench AntSoccer Arena Stitch.** We present the initial state for planning and each individual trajectories $\tau_{1:4}$ above. The start position of the ant is shown by the blue circle (bottom right) and the goal is to move the ball to the pink circle (upper right). We compositionally sample 4 trajectories (as shown from left to right), which will then be merged to form a long-horizon plan. The ball is highlighted with a small yellow circle. Our compositional sampling method effectively stitches two different types of trajectories and generalizes to more difficult tasks unseen in the training data.

# D Failure Mode Analysis

In this section, we present several failure cases of our method along with analyses and qualitative examples of these failure modes. Our proposed framework consists of two components: a generative planner and an inverse dynamics model. We outline and discuss several failure modes below.

## D.1 Infeasible Generated Plan

As the number of composed trajectories $K$ increases, our method may show inconsistencies in the trajectory plan, especially when planning in high dimensional state space. We observe that although the start and goal conditioning can consistently ensure that the composed plan begins at the initial state and terminates at the provided goal, the intermediate trajectories may include implausible transitions, such as trajectories that jump over walls, making the overall plan infeasible. However, empirically these failures usually occur at considerably higher $K$ as compared to baseline methods (e.g., GSC, see Table 1 and Table 2).

In Figure 15, we present a qualitative example of infeasible plan in OGBench `AntMaze Giant Stitch` when planning in 29D space (ant's $x$-$y$ position, joint positions, and joint velocity) and composing 9 trajectories. The original white walls are rendered transparent for better view of infeasible states. Though the generated compositional plan is mostly valid, the second trajectory $\tau_2$, as shown in yellow, is infeasible as it passes through walls. This is probably due to the suboptimal coordination among trajectories in the compositional sampling process.

As illustrated in the figure, certain trajectories (e.g., the neighboring blue, green, and red ones) barely proceeds, moving only 2-3 blocks. To bridge the resulting spatial gap, the intermediate yellow trajectory must span a much longer distance. However, the training data does not contain such long-horizon trajectories, making it difficult for a single trajectory to extend to such length, which

finally results in an o.o.d sample that goes through walls. We could potentially mitigate this issue with rejection or guided sampling techniques to select feasible candidate plans.

## D.2 Suboptimal Inverse Dynamics Model

In our experiments, we observe that even if CompDiffuser synthesizes a feasible trajectory for the agent to follow, the agent might not successfully reach the goal due to the error by the inverse dynamics model. For example, the agent might bump into walls due to unstable locomotion or get stuck in some local region, yet the synthesized plan is collision-free and coherent.

In implementation, we use a simple MLP to parameterize the inverse dynamics model and train it with a regression MSE loss. We believe that more optimized model architecture or specific finetuning might further boost the performance of the inverse dynamics model, hence boosting the overall performance of our method. We deem that out of the scope of this work.

In Figure 16, we present a qualitative example of failure due to the inverse dynamics model in OGBench `AntMaze Giant Stitch`. The planning is in 15D space (ant's *x-y* position and joint positions) and composes 3 trajectories. While we observe that the synthesized plan is feasible and successfully reaches the goal, the environment execution rollout fails during the yellow trajectory. In this instance, the ant agent fails to execute the right turn, loses track of subsequent subgoals, and becomes trapped in a local region. To address this issue, incorporating effective replanning strategies could help the agent recover (since the new plan will start from the current trapped state). In addition, we deem that employing more robust or specialized inverse dynamics models may further mitigate such failure scenarios.

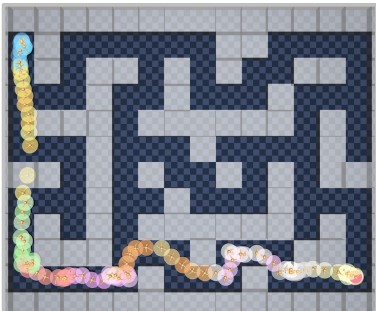

Infeasible Plan: Passing through Walls

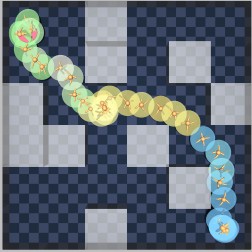

Feasible Plan

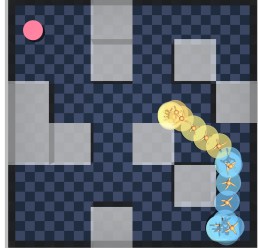

Rollout: Suboptimal Actions

Figure 15: **Failure Mode: Infeasible Plan.** We increase the transparency of the original white walls for better view of infeasible states. The start state is at the top left corner and the goal state is at the bottom right corner. Nine trajectories are composed, highlighted by different colors. The second trajectory $\tau_2$ (shown in yellow) is infeasible and passes through walls. This is probably due to the suboptimal coordination between trajectories in the compositional sampling process: the neighboring blue and green trajectories barely progress, leaving a long o.o.d gap for the yellow trajectory to fill.

Figure 16: **Failure Mode: Suboptimal Inverse Dynamics Actions.** We present a failure case of planning in 15D space (ant's *x-y* position and joint positions) in `AntMaze Medium Stitch`. Left: the compositional plan where three trajectories (represented in blue, yellow, and green) are composed. We sub-sample the plan to 36 states for visualization (the original plan is dense with over 300 states). Right: the corresponding environment execution rollout of the compositional plan. Though the synthesized plan is valid and successfully reaches the goal, the inverse dynamics model may fail, as illustrated on the right which gets stuck in a right turn and is unable to proceed. Further incorporating some effective replanning strategies or employing more robust inverse dynamics models could potentially mitigate these failure scenarios.

## D.3 Suboptimal Number of Composed Trajectories $K$

In our current implementation, the number of composed trajectories $K$ needs to be manually specified at test time. As shown by the quantitative results Table 8 and qualitative results Figure 10, a relatively larger $K$ does not significantly affect the model performance as the extra plan length will be consumed by staying or circulating within certain valid regions in the environment. However, an aggressively smaller $K$ might lead to failure plans – since the overall planning horizon becomes insufficient to cover the substantial spatial gap between the start and the goal.

In this section, we provide qualitative examples of infeasible plans due to impractical $K$. In Figure 17, we show each individual trajectory $\tau_k$ generated by our compositional sampling method when $K$ ranges from 3 to 6. Note that a feasible horizon to reach the goal from the start (bottom left) to the goal (upper right) requires composing at least 8 trajectories, i.e., $K = 8$.

Therefore, while the generated trajectories can begin at the start state and terminate at the goal state, the intermediate segments become disconnected since there are too few trajectories to bridge the significant spatial gap (given that the training data contains only short trajectories). Nonetheless, we observe that the overall flow of the trajectories is directed toward the goal, and as $K$ increases, the plan's structure gradually becomes more feasible.

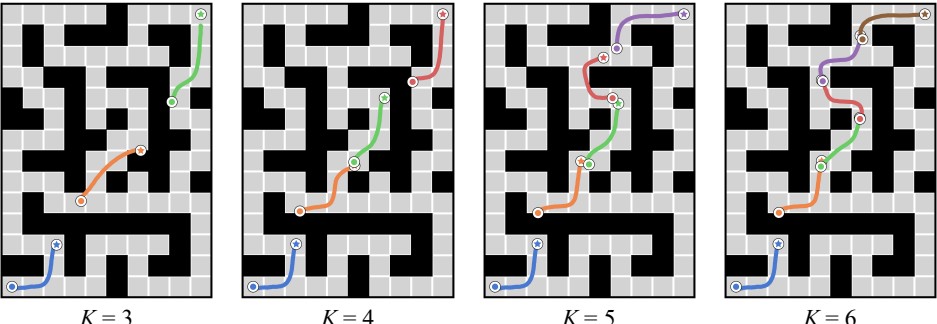

| $K = 3$ | $K = 4$ | $K = 5$ | $K = 6$ |

Figure 17: **Failure Model: Suboptimal Number of Composed Trajectories $K$.** Our method may generate infeasible plans if $K$, the number of composed trajectories, is set significantly below the required minimum. For example, in the planning task illustrated above, the start state is located in the bottom left corner while the goal state is at the upper right corner. A feasible plan for this task typically requires composing at least 8 trajectories (i.e., $K = 8$). We present qualitative examples of plans when $K$ is smaller than the minimum threshold. Though the first and last trajectory segments correctly attach to the start and goal states, the intermediate trajectories are disconnected due to the insufficient plan length. However, the overall plan still progresses towards the goal, which may provide useful guidance signals to the agent.

# E    Implementation Details

**Software:** The computation platform is installed with Ubuntu 20.04.6, Python 3.9.20, PyTorch 2.5.0.

**Hardware:** We use 1 NVIDIA GPU for each experiment. A GPU with 24GB memory is sufficient to train our models and it takes 1-2 days to train a model using a recent mid-level NVIDIA GPU.

## E.1    Our Conditional Diffusion Model

In this section, we introduce detailed implementation of our conditional diffusion model $\epsilon_\theta(\tau^t, t \mid \text{st\_cond}, \text{end\_cond})$.

**Model Inputs and Outputs.** Our diffusion model takes as input the noisy trajectory $\tau^t$, diffusion noise timestep $t$, and the start condition $\text{st\_cond}$ and end condition $\text{end\_cond}$ for $\tau^t$. $\text{st\_cond}$ can be either noisy chunk or start state $q_s$ and $\text{end\_cond}$ can be either noisy chunk or goal state $q_g$. We use the predicting $\tau^0$ formulation to implement this network, i.e., the network directly predicts the clean sample $\tau^0$.

**Implementation of Noisy Chunk Conditioning.** As described in Section 3, we only train *one* diffusion model $\epsilon_\theta$ that is able to condition on both the start state $q_s$, goal state $q_g$, and noisy chunk $\tau^t_{k-1}$, $\tau^t_{k+1}$, such that in test time we can use the proposed compositional sampling approach to construct long-horizon plans. This unified one model design eliminate the need for manual task subdivision (across multiple models) or reliance on predefined planning skeleton, thereby enabling holistic end-to-end planning.

In practice, we can implement the noisy neighbor conditioning $\mathcal{L}_{\text{nbr}}$ simply using the overlapping part between the two chunks instead of a full chunk $\tau_{k-1}/\tau_{k+1}$ conditioning, because the overlapping part is sufficient to ensure the connectivity between two adjacent trajectories. Such design further simplifies the training procedure: instead of dividing a training trajectory $\tau$ to $K$ chunks, we only need to sample a noisy sub-chunk from $\tau$ itself.

Specifically, assume $\tau^t$ is the noisy trajectory to be denoised and $\hat{\tau}^t$ is another independent noisy version of $\tau$ also at noisy timestep $t$. We denote the length of $\tau$ as $h$ and the length of the overlapped part as $h_\mathrm{o}$, then we can use $\hat{\tau}^t[0 : h_\mathrm{o}]$ and $\hat{\tau}^t[h - h_\mathrm{o} : h]$ as the noisy sub-chunks for st_cond and end_cond during training, respectively. Hence, when inference, the end_cond for $\tau_1$ can be set to $\hat{\tau}_2^t[0 : h_\mathrm{o}]$ (the front chunk of the second trajectory $\tau_2$) and the st_cond for $\tau_2$ can be set to $\hat{\tau}_2^t[0 : h_\mathrm{o}]$ (the tail chunk of the first trajectory $\tau_1$).

**Model Architecture.** For planning in 2D *x-y* space, we follow Decision Diffuser [1] (https://github.com/anuragajay/decision-diffuser/) and use a conditional U-Net as the denoiser network. For planning in high dimension state space, we use a DiT [55] based transformer [65] as the denoiser network (https://github.com/facebookresearch/DiT/).

**Training Pipeline.** We provide detailed hyperparameters for training our model on `PointMaze Giant Stitch` environment in Table 12. We do not apply any hyperparameter search or learning rate scheduler. Please refer to our codebase for more implementation details.

| Hyperparameters | Value |
|---|---|
| Horizon | 160 |
| Diffusion Time Step | 512 |
| Probability of Condition Dropout | 0.2 |
| Iterations | 1.2M |
| Batch Size | 128 |
| Optimizer | Adam |
| Learning Rate | 2e-4 |
| U-Net Base Dim | 128 |
| U-Net Encoder Dims | (128, 256, 512, 1024) |

Table 12: Hyperparameters for Training on `PointMaze Giant Stitch` environment.

**Inference Pipeline.** In Table 13, we present the single model horizon (length of individual $\tau_k$) and the inference-time number of composed trajectories $K$ corresponding to the reported results in Table 1, Table 2 and Table 3.

For each evaluation problem, we generate $B$ samples in a batch and we use a simple heuristic to select one sample as the output plan. Specifically, we compute the L2-distance of each overlapping parts in the generated trajectory segments $\tau_{1:K}$. The one with the smallest average distance will be adopted as the output plan, in the sense that a small distance in the overlapping parts indicates better coherency between adjacent trajectories. we deem that developing some more advanced inference-time methods with CompDiffuser may be an interesting future research direction, such as probability or density based plan selection methods or compositional sampling with flexible preference steering.

### E.2 Trajectory Merging

As introduced in Method Section 3 and Algorithm 2, the generated trajectories $\tau_{1:K}$ are mutually overlapped and we merge these $K$ trajectories to form a long-horizon compositional trajectory $\tau_\mathrm{comp}$. In this section, we describe the implementation of the exponential trajectory blending technique which we use for merging.

We directly leverage the classic exponential trajectory blending formulation. For simplicity, let $\tau_1$ and $\tau_2$ denote the trajectories to blend, $\tau_1[t]$ denote the $t$-th state in $\tau_1$, $t_\mathrm{start}$ and $t_\mathrm{end}$ denote the start and end index of the region to apply blending. Note that, in practice, we *only* blend the overlapped part between two adjacent trajectories. We provide the equation for exponential blending below,

$$\tau_\mathrm{comp}[t] = w(t) * \tau_1[t] + (1 - w(t)) * \tau_2[t], \quad \text{where } w(t) = \frac{e^{-\beta\left(\frac{t - t_\mathrm{start}}{t_\mathrm{end} - t_\mathrm{start}}\right)} - e^{-\beta}}{1 - e^{-\beta}} \quad (7)$$

We set $\beta = 2$ across all our experiments. In practice, various other trajectory blending techniques can also be directly applied, such as cosine blending and linear blending.

| Environment | Type | Size | Single Model Horizon | # of Composed Trajectories |
|---|---|---|---|---|
| PointMaze [21] | - | U-maze | 40 | 5 |
| | | Medium | 144 | 5 |
| | | Large | 192 | 5 |
| pointmaze | Stitch | Medium | 160 | 3 |
| | | Large | 160 | 5 |
| | | Giant | 160 | 8 |
| AntMaze | Stitch | Medium | 160 | 3 |
| | | Large | 160 | 6 |
| | | Giant | 160 | 9 |
| AntMaze | Explore | Medium | 192 | 5 |
| | | Large | 192 | 6 |
| AntSoccer | Stitch | Arena | 160 | 5 |
| | | Medium | 160 | 6 |
| HumanoidMaze | Stitch | Medium | 336 | 4 |
| | | Large | 336 | 6 |
| | | Giant | 336 | 11 |

Table 13: **Number of Composed Trajectories for Each Evaluation Environment.** Our diffusion models are trained with a short horizon as listed in the *Single Model Horizon* column. In test time, we compositionally generate multiple such short trajectories to enable trajectory stitching and construct plans of much longer horizon.

### E.3 Replanning

In this section, we describe the detailed implementation of replanning. While our method is designed to directly generate an end-to-end trajectory from the given start state to the goal state, replanning can be performed at any given timesteps during a rollout. Specifically, we initiate replanning if the agent loses track of the current subgoal, i.e., the L2 distance between the agent and the subgoal is larger than a threshold.

In a larger maze, the required number of composed trajectories, denoted as $K$, is usually large due to the distance between the start state and the goal state. However, if we keep replanning with a similar large $K$ even when the agent is already close to the goal, the generated trajectory might travel back and forth to consume the unnecessary intermediate length (see (2) and (3) in Figure 12), thus delaying the agent's progress toward the goal.

To address this, we use a receding scheme for $K$ to encourage faster convergence to the goal. Let $K$ denote the number of composed trajectories of the current plan (the plan that the agent is following), $h_{\text{comp}}$ denote the length of the current plan, $h_{\text{exe}}$ denote the length of the current plan that the agent already executes in the environment. The number of composed trajectories used for replanning $K_{\text{replan}}$ is given by

$$K_{\text{replan}} = \texttt{ceil}\left( \left(1 - \frac{\max(h_{\text{exe}} - \delta,\, 0)}{h_{\text{comp}}}\right) * K \right) \tag{8}$$

where $\delta$ is a hyper-parameter that controls the convergence speed to the goal, for example, $K_{\text{replan}}$ will decrease faster if $\delta$ is set to a small (or negative) number while $K_{\text{replan}}$ will decrease very slowly if $\delta$ is a large positive number.

## F   Baselines

We compare our approach with a wide variety of baselines, including diffusion-based trajectory planning algorithms, data augmentation based stitching algorithms, and goal-conditioned offline RL algorithms.

Particularly, we include the following methods:

- For generative planning methods, we include Decision Diffuser (DD) [1] for monolithic trajectory sampling and Generative Skill Chaining (GSC) [48] for trajectory stitching;

- For data augmentation based methods, we include stitching specific data augmentation [21] with RvS [14] and Decision Transformer (DT) [9]. Since the evaluation setting is identical, we directly adopt the reported numbers in the original paper [21].

- For offline reinforcement learning methods, we include goal-conditioned behavioral cloning (GCBC) [44, 20], goal-conditioned implicit V-learning (GCIVL) and Q-learning (GCIQL) [29], Quasimetric RL (QRL) [70], Contrastive RL (CRL) [15], and Hierarchical implicit Q-learning (HIQL) [52]. For these baselines, we follow the implementation setup established by OG-Bench [51] throughout our experiments.

We describe the implementation details of a few notable ones below.

**Generative Skill Chaining (GSC) [48].** GSC is a recent diffusion model-based skill stitching method. We directly adopt its original score-averaging based stitching algorithm and apply it in our tasks. Specifically, when composing $K$ trajectories $\tau_{1:K}$, GSC averages the scores of the overlapping segments between adjacent trajectories (in total $K-1$ overlapping segments in this case) prior to each denoising timestep. For a fair comparison, we re-use the same diffusion denoiser networks in CompDiffuser for every GSC experiment.

**Hierarchical Implicit Q-Learning (HIQL) [52].** HIQL is a recently proposed Q-learning based method that employs a hierarchical framework for training goal-conditioned RL agents. It learns a goal conditioned value function and uses it to learn feature representations, high-level policy, and low-level policy. We follow the original implementation of the method in OGBench [51].

**Goal-Conditioned Behavioral Cloning (GCBC) [44].** GCBC is a classic imitation learning-based method. In our experiments, GCBC trains an MLP that takes as input the observation state and a future goal state from the same offline trajectory and outputs a corresponding action for the agent. We use the same implementation as in OGBench [51].

