# OpenReview forum: "Generative Trajectory Stitching through Diffusion Composition"
_NeurIPS.cc/2025/Conference — NeurIPS 2025 spotlight_

### Official Review · Reviewer_qv69 · 2025-06-28

**Clarity:** 3
**Significance:** 3
**Originality:** 3
**Rating:** 5
**Confidence:** 4

**Summary:**

The authors present CompDiffuser a method for planning trajectories in a sampling driven generative manner based on diffusion. Long and potentially complex trajectories are decomposed into several shorter shorter trajectory segments to solve the overall motion planning problem. The conditional relationships between consecutive trajectory chunks are then learned using a bidirectional diffusion model. The proposed method enables both parallel generation of trajectory chunks and causal autoregressive generation and is shown to significantly outperform several relevant baselines.
The authors present an experimental evaluation of their approach both against multiple baselines and for several scenarios in simulation. Experimental simulation settings include 2d maze navigation problems and higher dimensional maze navigation settings with ant and humanoid embodiments where the method is shown to significantly outperform existing baselines.

**Questions:**

- Classical motion planning techniques in robotics such as probabilistic roadmaps/rapidly exploring random trees construct graphs or trees with nodes and connecting trajectory segments between sets of states. Could the proposed stitching techniques potentially be extended to construct not just a single multi-segment path, but a complete graph of segments to solve motion planning problems between sets of source/target locations as well?

**Ethical Concerns:**

["NO or VERY MINOR ethics concerns only"]

**Final Justification:**

Following the authors' rebuttal I do maintain my accept recommendation for this paper which I believe proposes a valuable new method with the potential to have impact in motion planning for robotics. I do maintain my suggestion to the authors to consider additional more challenging motion planning problems in the experimental evaluation, but believe that the existing experimental evaluation is sufficient to accept the work to this conference.

**Limitations:**

Yes

**Quality:**

3

**Strengths And Weaknesses:**

Strengths:

- The paper is clearly written and provides a good overview of the proposed method.
- The supplementary material is exemplary and extensive and covers important aspects such as the failure mode analysis.
- The paper tackles a crucial current shortfall of diffusion based methods in motion planning and appears to make significant advances over baseline state of the art methods.

Weaknesses:

- The current experimental evaluation is still only based on relatively limited simulation problems. It would be interesting to see examples of challenging planning problems where the method still fails and a real world implementation in the robotics context would also be interesting to see in future work.

---

> ### Author Rebuttal · Authors · 2025-07-31
>
> We thank Reviewer qv69 for the positive and careful review. Please see our response to the listed concerns below.
>
> > W1-1. The current experimental evaluation is still only based on relatively limited simulation problems. It would be interesting to see examples of challenging planning problems where the method still fails.
>
>
> Thank you for your valuable comments.
>
> In our experimental setup, we include tasks of varying difficulty levels — from relatively simple point maze navigation tasks to very challenging tasks such as AntSoccer.
>
> For example, in the PointMaze-Large-Stitch task, our method achieves a 100% success rate. In contrast, in the challenging AntSoccer-Medium-Stitch task, although our method outperforms all baselines, the absolute success rate remains relatively low (17%), highlighting significant room for future improvement.
>
>
>
> > W1-2. A real world implementation in the robotics context would also be interesting to see in future work.
>
>
> Thank you for the valuable suggestions.
>
> The focus of our work is to propose a core algorithm for trajectory stitching based on compositional diffusion models. Empirically, we have conducted extensive experiments on various simulation environments (e.g., different task objectives, data quality, different embodiments and environment sizes) and have shown the efficacy of the proposed method.
>
> Since the proposed method makes no privileged knowledge about the simulation environments or tasks, it is applicable to a wide-range of real-world trajectory planning problems, which will be our future research focus.
>
> Additionally, during the rebuttal phase, we conduct an experiment on an OGBench Cube task, 'cube-single-play-v0'. Since this dataset is of `play` data type and not designed for trajectry stitching, we segment each long play-style trajectory in the dataset to multiple short segments and train CompDiffuser on these short segments.
>
> Specifically, the dataset contains 1000 trajectories of length 1000 and we train CompDiffuser only on short segments of length 48.
> In test time, since a plan of length 48 is not sufficient to complete a task, we compose three models and use the sampling scheme in Algorithm 2 to sample a plan of length 112 (i.e., $\tau_{\text{comp}}$). Similarly, we train an inverse dynamic model to output actions, such that the plan can be followed by the robot.
>
>
>
> Please see the results below. Results of the baselines are from the OGbench paper.
>
>
> | Env | GCBC | GCIVL | GCIQL | QRL | CRL | HIQL | Ours |
> | - | - | - | - | - | - | - | - |
> | `cube-single-play-v0` | 6±2 | 53±4 | 68±6 | 5±1 | 19±2 | 15±3 | 92±4 |
>
>
>
>
> > Q1. Classical motion planning techniques in robotics such as probabilistic roadmaps/rapidly exploring random trees construct graphs or trees with nodes and connecting trajectory segments between sets of states. Could the proposed stitching techniques potentially be extended to construct not just a single multi-segment path, but a complete graph of segments to solve motion planning problems between sets of source/target locations as well?
>
>
> Thank you for the insightful comments.
>
> We agree that extending the proposed method to solve motion planning problems between sets of source/target locations is a promising research direction, and we will explore along this line in our future work.
>
> Our current method generates a single long-horizon trajectory by compositionally denoise each segment $\tau_{1:N}$. These segments, namely $\tau_{1:N}$, are sequentially arranged in a line to form a continuous multi-segment path.
>
> To extend this to a graph of segments, we can first organize segments $\tau_{1:M}$ in a graph like structure (each segment can be viewed as an edge and $M$ is the number of segments in the graph).
> We can then denoise the whole graph in a similar fashion to Algorithm 2.
> Specifically, if the segment starts from a source location, we use Line 6 of Algorithm 2; if it is an intermediate segment, we use Line 9; if the segment terminates at a target location, we use Line 12.

---

> > ### Comment · Reviewer_qv69 · 2025-08-05
> > **Rebuttal response**
> >
> > I would like to thank the authors for their rebuttal response which clarified some of the questions I had. I enjoyed reading this paper and am maintaining my accept recommendation.

---

> ### Author Response · Authors · 2025-08-05
> **Thank you**
>
> Dear Reviewer qv69,
>
> We sincerely appreciate your thoughtful and detailed feedback, as well as your positive evaluation of our manuscript. Thank you once again for your time and effort in reviewing our work.
>
> Best,
> Paper Authors

---

### Official Review · Reviewer_AQzW · 2025-06-30

**Clarity:** 3
**Significance:** 3
**Originality:** 3
**Rating:** 5
**Confidence:** 4

**Summary:**

The paper presents a novel approach to planning for long-horizon tasks, called CompDiffuser. This method involves breaking down trajectories into shorter segments with overlapping parts, and then training diffusion models to learn the distribution of consecutive states conditioned on the start and end states of neighboring segments.

At test time, given a starting state and a goal state, CompDiffuser generates a plan by combining these segments in an autoregressive or parallel manner, resulting in a physically plausible plan. The plan is then executed using a learned inverse dynamic policy.

The paper provides empirical results, primarily focusing on navigation tasks with various embodiments (such as pointmass, ant, and humanoid) and different mazes. These results are compared to other state-of-the-art methods in goal-conditioned offline RL and diffusion-based approaches

**Questions:**

see comments above

**Ethical Concerns:**

["NO or VERY MINOR ethics concerns only"]

**Final Justification:**

the authors address my concerns, I raised my score

**Limitations:**

yes

**Paper Formatting Concerns:**

no issues

**Quality:**

3

**Strengths And Weaknesses:**

Strengths:


- The paper is well-structured, clearly motivated, and presents a straightforward methodology.
- The proposed CompDiffuser algorithm demonstrates good performance in stitching trajectories, outperforming baselines in most domains, particularly in longer horizon tasks (e.g., maze giant). Visualizations confirm that the generated plans are physically plausible.
- The paper provides comprehensive ablation studies on various aspects, including chunk size, state dimensionality, parallel vs. autoregressive sampling, and replanning procedures.


Weaknesses:

- A significant weakness of the paper is the lack of detailed implementation specifics, such as: Architecture details, the way they condition on overlapping parts, Network capacity (total number of parameters needed), any data normalization techniques, any weighting used in the diffusion loss?
- The usage of the generative skill chaining baseline is unclear; more details, equations, or explanations would be helpful to understand its application in this context.

- The trajectory merging approach appears to assume linear interpolation between produced trajectories. However, this might not be suitable for cases where two states are separated by obstacles (e.g., walls). Clarification on this aspect is necessary.

- Although the method seems generalizable, the empirical investigation primarily focuses on navigation tasks and planning with xy positions, leaving questions about its applicability to other domains, such as manipulation or locomotion. Could you provide results in cube domains in OGbench for example (single and double) ?
- Why in most of the results except in antmaze, the authors ignore the qvel part of the states, how bad the prediction would be if we predict the full state instead only the positions ?
- the method seems to rely on high quality data, expert demonstration. when trained on navigate data, the method performance decreases

---

> ### Author Rebuttal · Authors · 2025-07-31
>
> We thank Reviewer AQzW for the positive and careful review. Please see our response to the listed concerns below.
>
> > W1: A significant weakness of the paper is the lack of detailed implementation specifics, such as: Architecture details, the way they condition on overlapping parts, Network capacity (total number of parameters needed), any data normalization techniques, any weighting used in the diffusion loss?
>
>
> Thank you for your valuable comments. In our original manuscript, we described our implementation in Appendix E, including model inputs/outputs, noisy (overlapping) chunk conditioning, and model architectures, and we will make sure to add all the mentioned implementation specifics in our final manuscript.
>
> Please see the additional implementation details below:
>
> **Architecture details:**
> For planning in 2D x-y space, we follow Decision Diffuser and use a conditional U-Net as the denoiser network. The main modification we made it to concatenate the encoded latents of the noising neighbor trajectories to the conditional latents for denoising.
>
> For planning in high dimension state space (Table 3 and Table 4), we use a DiT based transformer as the denoiser network. We stack 4 states to construct a transformer input token. (For example, there will be 40 tokens for a trajectory with 160 states.) The latent dimension and depth of the transformer are 768 and 16.
>
>
> **Network capacity** (total number of parameters needed):
> For experiment results for Table 1, we have the parameter count as follows:
> PointMaze from [1]
> | Env | DD | Ours |
> | - | - | - |
> | U-Maze |  35.7 M  | 59.8 M |
> | Medium |  35.7 M  | 61.1 M |
> | Large |   35.7 M  |  61.4 M |
>
> OGBench PointMaze
> | Env | DD | Ours |
> | - | - | - |
> | Medium |  63.8 M  | 92.9 M |
> | Large |   63.8 M  |  92.9 M |
> | Giant |   63.8 M  | 92.9  M |
>
>
> OGBench AntMaze
> | Env | Ours |
> | - | - |
> | Medium  | 92.9 M |
> | Large   |  92.9 M |
> | Giant   | 92.9  M |
>
> OGBench HumanoidMaze
> | Env | Ours |
> | - | - |
> | Medium  | 95.8 M |
> | Large   | 95.8 M |
> | Giant   | 95.8  M |
>
>
> **Data Normalization:**
> We use the standard min-max normalization to normalize the training data to range (-1, 1).
>
> **loss weighting:**
> We do not use any weighting in the diffusion loss.
>
>
>
>
>
>
> > W2. The usage of the generative skill chaining baseline is unclear; more details, equations, or explanations would be helpful to understand its application in this context.
>
>
>
> Generative Skill Baseline (GSC) is a compositional inference method that averages the scores of the overlapping parts between two segments throughout the denoising process. In contrast, CompDiffuser's compositional inference directly conditions the denoising score function on the scores of the neighboring segments.
>
> Mathematically at denoising step $t$, with score of the neighboring segments as $\epsilon_{k-1}$ and $\epsilon_{k+1}$, with an overlapping region length $H_o$, we have the score of the overlapping region as
>
> $$\epsilon_k[:H_o] = (\epsilon_{k-1}[-H_o:] + \epsilon_k[:H_o])/2$$
> $$\epsilon_k[-H_o:] = (\epsilon_{k}[-H_o:] + \epsilon_{k+1}[:H_o])/2$$
>
>
> Both GSC and CompDiffuser can enable cross trajectory information exchange during denoising, hence promoting stitching capability.
> GSC enables such information exchange via simply replacing the score of the overlapping part between two segments by the average score of the corresponding parts from the two segments; while in CompDiffuser, we propose a more unified stitching framework based on the proposed autoregressive sampling method and bi-directional information passing (Algorithm 2).
>
> In Appendix F of our original manuscript, we described how we apply GSC in our benchmark and we will make sure to include the detailed introduction above in our updated manuscript.
>
>
>
>
> > W3. Trajectory merging might not be suitable for cases where two states are separated by obstacles (e.g., walls). Clarification on this aspect is necessary.
>
>
> We do not use trajectory merging to avoid obstacles, and CompDiffuser as a planner is tasked to sample valid trajectories that satisfy the obstacle contraints.
>
> Specifically, we propose CompDiffuser which stitches trajectories by generating a set of feasible and consecutive segments $\tau_{1:N}$.
> Then, trajectory merging (detailed in Appendix E.2) is only used to merge these compositionally-generated segments $\tau_{1:N}$ and form a long-horizon trajectory $\tau_{comp}$ for the agent to execute.
>
> As in our experiments (e.g., Table 1 and Figure 10/11), we also find that CompDiffuser is able to generate states that travel in valid areas and hence can result in a successful $\tau_{comp}$.
>
> Additionally, in some cases when the planner does fail (two states are separated by obstacles), other techniques can also be employed to refine the generated trajectory, such as perform replanning or leverage some sampling-based planning method to refine the local collision region.
>
>
>
>
>
> > W4. the empirical investigation primarily focuses on navigation tasks and planning with xy positions, leaving questions about its applicability to other domains, such as manipulation or locomotion. Could you provide results in cube domains in OGbench for example (single and double) ?
>
> Thank you for the insightful suggestions.
>
> In this paper, we aim to study the trajectory stitching problem and hence, in our original manuscript, we focus on evaluating our method on stitching-specific benchmarks, such as various maze stitching tasks and ant soccer stitching tasks in OGBench, along with multiple stitching datasets from [1].
>
> During the rebuttal period, due to the time and computation constraints, we conduct additional experiments on the `cube-single-play-v0` env.
> Since this `play` env is not designed for trajectory stitching, we segment each long play-style trajectory in the dataset to multiple short segments, and train CompDiffuser on these short segments.
> That is, the dataset contains 1000 trajectories of length 1000 and we train CompDiffuser only on short segments of length 48.
> In test time, since a plan of length 48 is not sufficient to complete a task, the planner must stitch multiple trajectories to construct a successful plan.
> Specifically, we compose three models and use the sampling scheme in Algorithm 2 to sample a plan of length 112 (i.e., $\tau_{\text{comp}}$). Similarly, we train an inverse dynamic model to output actions, such that the plan can be followed by the robot.
>
>
> Please see the results below. Results of the baselines are from the OGBench paper.
>
>
> | Env | GCBC | GCIVL | GCIQL | QRL | CRL | HIQL | Ours |
> | - | - | - | - | - | - | - | - |
> | `cube-single-play-v0` | 6 ± 2 | 53 ± 4 |68 ± 6 | 5 ± 1 | 19 ± 2 | 15 ± 3 | 92±4 |
>
>
>
>
>
>
>
>
> > W5. Why in most of the results except in antmaze, the authors ignore the qvel part of the states, how bad the prediction would be if we predict the full state instead only the positions?
>
>
> The primary objective of a trajectory stitching algorithm is to find sequence of subgoals that are reachable from the start state and allows us to progress towards the goal state. Thus, only the xy or xy with joint states is sufficient to determine a feasible subgoal.
>
> In our original submission, we presented an ablation study on the prediction dimension in Table 4 where we compare the success rates of predicting xy (2D), xy with joint states (17D), and xy with joint states and joint velocities (full state, 29D).
> Quantitatively, we observe that the performance loss is minor in smaller mazes such as Medium and Large, but we do see a drop in success rate in the most challenging Giant maze, likely due to modeling complexity of the full state planning.
>
> For further reference, we conduct additional experiments on OGBench `pointMaze-stitch` environments, where CompDiffuser predicts full state, both x-y positions and velocities (4D).
> Please see the success rate comparison below.
>
> | Env | qpos | qpos+qvel |
> | - | - | - |
> | `pointmaze-medium-stitch-v0` | 100±0 | 100±0  |
> | `pointmaze-large-stitch-v0` | 100±0  | 100±0 |
> | `pointmaze-giant-stitch-v0` |  68±3   | 72±4 |
>
>
> For the low-dimensional setup of `pointmaze-giant-stitch-v0`, we can see comparable performance when predicting full state (qpos+qvel) as compared to xy (qpos) positions only. This is in contrast with the findings from the experiment with `antmaze-giant-stitch-v0` environment.
> We speculate that the decreased results on the antmaze is due to the insufficient neural network capacity (higher modeling complexity to plan in a larger state space) and limited training data (while planning in antmaze is more difficult, the size of the antmaze dataset is identical to the size of the pointmaze dataset).
>
>
>
>
>
>
>
>
>
>
> > W6. The method seems to rely on high quality data, expert demonstration. when trained on navigate data, the method performance decreases.
>
>
> In our experiments, we evaluated CompDiffuser on various data qualities and the results demonstrated that CompDiffuser is robust to data quality.
>
> As in Table 2, our method is able to outperform all the baselines on extremely low-quality datasets: the OGBench *Explore* datasets.
> (The data quality of these datasets is much lower than the OGBench *navigate* datasets.)
> The collection policy for the *Explore* datasets will randomly re-sample a new moving direction after every 10 steps and contain a large amount of action noise, making it very challenging to learn an effective policy (See Figure 9 for qualitative dataset examples).
>
> Our method can also surpass baselines when the data are of relatively higher quality, for example, as in the OGBench *Stitch* datasets (see Table 1 and 2). These datasets are of relatively higher qualitiy than *Explore*, where the data collection policy is a low-level directional policy trained via SAC with a high-level waypoint controller.
>
>
> [1] Closing the Gap between TD Learning and Supervised Learning - A Generalisation Point of View.

---

> > ### Author Response · Authors · 2025-08-05
> > **Official Comment by Authors**
> >
> > Dear Reviewer AQzW,
> >
> > We deeply appreciate the time and effort you have dedicated to reviewing our manuscript. We understand that this may be a particularly busy period, but we would like to respectfully follow up regarding our responses to your previous comments and suggestions. We are glad to provide any further clarification -- please let us know if you have additional concerns.
> >
> > Thank you again for your valuable input in reviewing our work, and we look forward to hearing from you soon.
> >
> > Best,
> > Paper Authors

---

### Official Review · Reviewer_a2a3 · 2025-07-01

**Clarity:** 2
**Significance:** 3
**Originality:** 3
**Rating:** 4
**Confidence:** 3

**Summary:**

This article introduces a method for generative planning using stitching. Their method denoises short segments conditioned on the noisy states of their neighbors, to form overall long coherent trajectories. They introduce two flavors of this method, one of which only conditions on the previous noise level, and another which takes the current noise level for one neighbor. The paper presents extensive evaluation on maze-type benchmarks as bundled in the OGBench benchmark and others, and demonstrate an advantage over various baselines.

**Questions:**

In the algorithm, it states that "clean trajectories" are sampled from the dataset, while in the experiment description it is stated that the experment is done on benchmarks with short segments of trajectories. Are they full trajectories or short segments? If they are short segments, how are these connected together with their neighbors, as is required for the training algorithm? I.e., to train the diffusion model, the two neighbors on either side are required inputs to the model.

**Ethical Concerns:**

["NO or VERY MINOR ethics concerns only"]

**Final Justification:**

The authors have adequately answered the comments and concerns of my review, but the rebuttal and other reviews have not moved my score into the next bracket.

**Limitations:**

yes

**Quality:**

3

**Strengths And Weaknesses:**

Strenghts:
- The paper is nice to read and the method appears elegant and fits the problem well
- There is an extensive number of baselines and experiments presented in the paper
- Effective learning based only on chunks that generalizes to longer horizons is an important task
Weaknesses:
- While the baselines are plentiful and there are a lot of experiments, it is not made entirely clear how appropriate the baselines actually are, and while there are a lot of experiments, there is not a lot of variation in task type. I will address each of these concern in a bit more detail.
Baselines:
- In appendix B.3, they compare against "monolithic", which is much faster, they say because it is a much smaller architecture. This seems like a strange comparison: it is not apples to apples, and therefore not very useful. As far as I see it, their parralel "chunked" attention is effectively monolithic attention with masking of all but three chunks, so, it should be possible for their method to be at least as fast or faster in like for like scenarios. This would indicate the overall model for the monolithic case is smaller, making the comparison invalid.
- Continuing on this same topic, the parameter count / model size is not really well presented, but seems like an important thing to report since they give such strong performance against baselines. This raises the question whether the authors simply train much bigger networks than their competitors and this is why they get such a big advantage. It is more useful to the reader if they can compare like for like. In Table 12 there are some hyperparameters reported, but it is not possible to understand overall parameter count from that, and it is not possible to understand how it compares to DD.
- In the checklist, authors state that their experiments are reproducible since they "describe it in Appendix E and they release their code on acceptance". In appendix E, the authors only describe the setup partially (only for `pointmaze giant`), and refer the reader to "refer to our codebase for details", which is not provided, making it impossible to judge any of these details. The authors should have an appendix with experimental settings for each experiment, and list total number of hyperparameters for each model, as well as how it compares to baselines.
- Something that is not made clear in the paper is whether baselines are trained on snippets as well, and then required to produce full trajectories at inference time? If this is the case that also seems like a comparison of limited use, since those methods were not developed to perform well on these tasks. Can the authors comment?
Experiments:
- The experimental evaluation is limited to maze-like RL benchmarks, especially given the baselines such as GSC, it would be ineresting to see performance on such robot manipulation tasks as well.

---

> ### Author Rebuttal · Authors · 2025-07-31
>
> We thank Reviewer a2a3 for the positive and careful review. Please see our response to each listed concern below.
>
>
> #### Baselines:
>
> > W1&W2. The inference time in appendix B.3 would indicate the overall model for the monolithic case is smaller, making the comparison invalid. Continuing on this same topic, the parameter count / model size is not really well presented, This raises the question whether the authors simply train much bigger networks than their competitors and this is why they get such a big advantage. In Table 12 there are some hyperparameters reported, but it is not possible to understand overall parameter count from that, and it is not possible to understand how it compares to DD.
>
>
>
> Thank you for your valuable comments. Please see our response to the model sizes and inference time below. In our final manuscript, we also will make sure to include all the mentioned implementation details.
>
>
> **Model Size.**
> For the result shown in Table 1, the denoiser model of DD is indeed smaller than ours.
> However, in our implementation, we already matched the architecture, depth, and feature dimension of the main denoiser network for DD and ours.
> The additional parameters come from the neighbor trajectory encoding networks (which encode $\tau_{k-1}$ and $\tau_{k+1}$ to latents) and MLPs for handling the resulting condition features of higher dimension (since these two neighbor trajectory latents are also used to condition the denoising network).
>
> Please see the parameter count of the networks below:
>
> PointMaze from [1]
> | Env | DD | Ours |
> | - | - | - |
> | U-Maze |  35.7 M  | 59.8 M |
> | Medium |  35.7 M  | 61.1 M |
> | Large |   35.7 M  |  61.4 M |
>
> OGBench PointMaze (networks used for reporting the sampling time in Table on Page 20)
> | Env | DD | Ours |
> | - | - | - |
> | Medium |  63.8 M  | 92.9 M |
> | Large |   63.8 M  |  92.9 M |
> | Giant |   63.8 M  | 92.9  M |
>
> For our method, we use the same main denoiser network across these mazes. However, since the trajectory lengths are different for mazes of different sizes in PointMaze[1], the size of the encoder networks for neighbor trajectories (i.e. $\tau_{k-1}$, $\tau_{k+1}$) also varies, resulting in a minor difference in model sizes for our method. We will make sure to enclose detailed model sizes in our updated manuscript accordingly.
>
>
> **Inference Time.**
> We agree that given a similar network, the parallel denoising scheme should yield a similar inference time as DD. We have also discussed the reasons for the time difference in Appendix B.3 (Line 664-671) in our original submission. We summarize the reasons below:
>     (1) DD's denoiser network is indeed smaller;
>     (2) Our method requires encoding two noisy trajectories into latent representations, which are then used to condition the main denoiser network. This encoding step cannot be performed in parallel with the forward pass of the denoiser, resulting in additional latency.
>     (3) Parallel sampling increases the inference batch size. In the parallel sampling scheme, all trajectories $\tau_{1:N}$ are stacked to one batch. And as described in Appendix E.1 (Line 822-831), we generate $B$ (set to 10) samples in a batch and we compute the L2-distance of each overlapping parts to select the best plan (the one with the smallest distance). As a result, the batch size increases to $K \times B$, which will increase sampling time.
>     (4) The overhead of trajectory merging. The construction of the final plan involves merging multiple trajectory segments, which might introduce further overhead.
>
>
>
> Additionally, we further increase the parameters of DD and conduct additional experiments for fairer comparison. To better reflect the planning performance, we show the results of DD and Ours without replanning.
>
>
> In the table below, we present results on PointMaze from [1], where we increase the base feature dimension from 96 to 128, such that DD's network is in similar size as ours.
> | Env | DD (35.7M) | DD (63.8M) | Ours (~60M) |
> | - | - | - | - |
> | U-Maze |  0±0  | 0±0 | 100±0 |
> | Medium |  30±1  | 16±0 | 100±0 |
> | Large |   0±0  | 28±0 | 100±0 |
>
>
> In the table below, we present results on OGBench PointMaze Stitch, where we increase the base feature dimension from 128 to 160, such that DD's network is in similar size as ours.
>
> | Env |DD (63.8M) | DD (99.0M) | Ours |
> | - | - | - | - |
> | Medium |  80±0  | 55±1 | 100±0 |
> | Large |  40±0 | 39±0 | 100±0 |
> | Giant |   0±0  | 0±0  | 53±6  |
>
> In these datasets, while the two DD models achieve similar performance on Large and Giant mazes, the smaller DD model even yield better performance than larger model in the simple Medium maze. We deem that this is likely due to overfitting in the larger model.
>
> [1] Closing the Gap between TD Learning and Supervised Learning - A Generalisation Point of View.
>
>
>
>
>
>
>
> > W3. In the checklist, authors state that their experiments are reproducible since they "describe it in Appendix E and they release their code on acceptance". In appendix E, the authors only describe the setup partially (only for pointmaze giant), and refer the reader to "refer to our codebase for details", which is not provided, making it impossible to judge any of these details. The authors should have an appendix with experimental settings for each experiment, and list total number of hyperparameters for each model, as well as how it compares to baselines.
>
>
>
> Thank you for the valuable suggestions.
> We will make sure to include an appendix section with complete experimental settings for each experiment in our updated manuscript. We also promise to release the code for reproducibility. (Due to the character limits, we are not able to present all the detailed experimental settings in the response.)
>
> The hyperparameters of our method is actually very similar across different experiments.
> In fact, the hyperparameters (e.g., training dynamics, model architectures) is identical for OGBench PointMaze and AntMaze as shown in Table 12.
>
> Since the trajectories in the humanoidmaze dataset is much longer (humanoidmaze 400 vs antmaze 200), we update several hyperparameters to fit the longer trajectories. Please see the table below:
>
> |Hyperparameters | Value |
> | - | - |
> |Horizon | 336 |
> |Diffusion Time Step | 1000 |
> |Probability of Condition Dropout | 0.2 |
> |Iterations | 1.2M |
> |Batch Size | 192 |
> |Optimizer | Adam |
> |Learning Rate | 2e-4 |
> |U-Net Base Dim | 128 |
> |U-Net Encoder Dims | (128, 256, 512, 1024) |
>
>
>
>
>
> > W4. Something that is not made clear in the paper is whether baselines are trained on snippets as well, and then required to produce full trajectories at inference time? If this is the case that also seems like a comparison of limited use, since those methods were not developed to perform well on these tasks. Can the authors comment?
>
>
>
> Yes. The baselines are also trained on the snippets (which are short trajectories with respect to the horizon of the test-time task). For each experiment, we use the same training dataset for all methods.
>
> Many of our baselines do process trajectory stitching ability. For example, RL methods can inherit the stitching ability through their formulation such as value propagation; while GSC is specifically designed for stitching tasks.
> While monolithic model (Diffuser[1], Decision Diffuser[2]) is not specifically designed for stitching, it is also shown to exhibit the ability of length generalization, potentially due to the generalizability of diffusion model itself and the neural network (convolutional U-Net). Our goal to add DD as a baseline is to show that our compositional sampling method is able to offer significant advantage on stitching and length generalization compared to a monolithic model method.
>
>
>
>
> #### Experiments:
>
>
> > W5. The experimental evaluation is limited to maze-like RL benchmarks, especially given the baselines such as GSC, it would be ineresting to see performance on such robot manipulation tasks as well.
>
>
>
>
> Thank you for the valuable comments. We agree that extending the proposed method to robot manipulation tasks is an interesting direction. During the rebuttal period, we also conduct additional experiments on the OGBench `cube-single-play-v0` environments and validate that our method can also work well on robot manipulation tasks. Please see the results below.
>
> | Env | GCBC | GCIVL | GCIQL | QRL | CRL | HIQL | Ours |
> | - | - | - | - | - | - | - | - |
> | `cube-single-play-v0` | 6 ± 2 | 53 ± 4 |68 ± 6 | 5 ± 1 | 19 ± 2 | 15 ± 3 | 92±4 |
>
>
>
>
>
>
>
> > Q1. Are the sampled "clean trajectories" in Algorithm 1 full trajectories or short segments? If they are short segments, how are these connected together with their neighbors?
>
>
> In Line 3 of Algorithm 1, we wrote 'sample clean trajectory from dataset'. The 'clean trajectory' here refer to a trajectory without any added noise (hence in diffusion noise level 0).
> This 'clean trajectory' can be considered as a 'full trajectory' in the dataset. However, to effectively evaluate stitching ability, the dataset only contains 'short' trajectories ('short' here is relative to the horizon of the test-time tasks).
>
> Then we can obtain multiple overlapping short segments from the 'full trajectory' from the dataset using Line 4 in Algorithm 1. Specifically, to obtain the neighbors for training, we further divide the sampled short segment $\tau_0$ to $K$ overlapped pieces. Since these $K$ pieces are from one consecutive trajectory, we can obtain valid data to train the model. For clarity, we will update our manuscript regarding the training data accordingly.

---

> ### Comment · Reviewer_a2a3 · 2025-08-01
>
> The authors have adequately answered my questions and concerns, but I will maintain my score.

---

> ### Author Response · Authors · 2025-08-05
> **Thank you**
>
> Dear Reviewer a2a3,
>
> We sincerely appreciate your thoughtful and detailed feedback, as well as your positive evaluation of our manuscript. Thank you once again for your time and effort in reviewing our work.
>
> Best,
> Paper Authors

---

### Official Review · Reviewer_yhYf · 2025-07-02

**Clarity:** 4
**Significance:** 4
**Originality:** 4
**Rating:** 5
**Confidence:** 4

**Summary:**

Briefly summarize the paper and its contributions. This is not the place to critique the paper; the authors should generally agree with a well-written summary. This is also not the place to paste the abstract—please provide the summary in your own understanding after reading.

This paper considers the problem of building compositionally generalizable diffusion-based planners, i.e. models that can diffuse long sequences of actions (and/or states), without being bound to a fixed plan length. While prior work on diffusion-based planners (e.g. Diffuser or Decision Diffuser) did allow for generating multiple fixed-size plans in sequence, they struggle to generalize to sequences of much longer sequences than ones used in training. One of the challenges is ensuring a smooth and physically feasible transition between denoised sub-plans.

This work proposes CompDiffuser, which addresses this by denoising several sub-plans at the same time, with each sub-plan being conditioned on its adjacent sub-plans. In addition, the sub-plans are chosen to be overlapping. This enables the model to sample sub-plans that can actually be stitched together more easily.

Concretely, the trajectories are factorized as:
$$
p_\theta(\tau \mid q_s, q_g) \;\propto\;
p_1\bigl(\tau_1 \mid q_s, \tau_2\bigr)\,
p_K\bigl(\tau_K \mid \tau_{K-1}, q_g\bigr)\,
\prod_{k=2}^{K-1}
p_k\bigl(\tau_k \mid \tau_{k-1}, \tau_{k+1}\bigr).
$$

Where $q_s$ and $q_g$ are the start and goal states, respectively.

These conditional distributions are modeled using a joint diffusion model. At training time, the noise prediction network (in the DDPM sense) for sub-trajectory $\tau_k$ at noise level $t$ (denoted $\tau_k^t$) takes as input $t, \tau_k^t, \tau_{k-1}^t$ and $\tau_{k+1}^t$, i.e. it is conditioned on the adjacent sub-trajectories at the same noise level. There is also a loss term for conditioning the first and last plans on the start and goal states, respectively. The fact that adjacent chunks influence each other during sampling aims to enable the model to make the sub-plans compatible “by construction”, allowing them to be easily stitched together at the end of sampling.

The authors propose two alternatives for sampling:

- **Parallel sampling**: the noise predictor at $\tau_k^t$ conditions on $\tau_{k-1}^t$ and $\tau_{k+1}^t$, as in training. This means that the samples at noise level $t-1$ only depend on samples at noise level $t$, meaning the denoising steps can be run in parallel.
- **Autoregressive sampling**: the noise predictor at $\tau_k^t$ conditions on $\tau_{k-1}^{t-1}$ and $\tau_{k+1}^t$ (i.e. the sample for the previous trajectory is fed in at noise level $t-1$). Now one needs to do each denoising step sequentially (across time) at each noise level. The authors claim this enables stronger coordination across sub-plans, and find that autoregressive sampling has better empirical performance.

Experiments focus on validating the empirical performance of CompDiffuser and studying how its performance varies with respect to state dimensionality, number of composed trajectories, sampling schemes and replanning.

Evaluation covers PointMaze stitching datasets and OGBench locomotion tasks (AntMaze, HumanoidMaze, AntSoccer), comparing CompDiffuser against diffusion planners (Decision Diffuser, GSC), sequence-modeling methods (RvS, DT), and offline RL algorithms (GCBC, GCIVL, GCIQL, QRL, CRL, HIQL). The training data for these environments is constrained to shorter trajectories, as opposed to whole plans, producing a setting where models need to compositionally generalize in order to solve the task. CompDiffuser consistently outperforms these baselines on PointMaze and high-dimensional stitching tasks, demonstrating its ability to maintain feasibility and goal reachability under varied conditions.

Ablations show autoregressive sampling yields higher success than parallel sampling; replanning offers robustness gains primarily in the most complex (Giant) mazes; and performance remains stable across segment counts and inverse-dynamics configurations but degrades as planning dimensionality increases.

**Questions:**

- I think one interesting statistic about planning methods is the size of the largest problem they can solve reliably. In the setting of PointMaze environments, this could be measured along the lines of “how large does the maze have to be for the average success rate to drop below 95%”. Do the authors have a sense of, in the setting of Table 1, for how large a horizon does CompDiffuser’s performance start degrading, if any? We see that from the Large maze to the Giant maze in OGBench there is a significant drop; I wonder at what problem size the degradation starts.
- On a similar vein, do the authors have a sense of how they would build a CompDiffuser to achieve 100% accuracy in the Giant maze? Is it a matter of simply collecting more data? Or is it potentially a fundamental limitation of the method due to accumulating errors throughout the autoregressive sampling process at longer horizons?
- A more general instance of the question above: what are the main variables with respect to which the generalization ability of the method scales? (Potential variables could be amount of data, ratio of training trajectory length to evaluation trajectory length, computation budget for denoising during the forward pass, replanning budget, etc).
- In appendix, the authors mention that CompDiffuser’s autoregressive performance is robust to whether one starts from the starting position (hence working “forward” to the goal position) or from the goal position (hence walking “backward” to the starting position). However, the locomotion problem here is in a sense very symmetric, in the sense that a valid plan from point A to point B can be inverted to obtain a valid plan from B to A. Do the authors expect the method to be robust to sampling direction in more general settings, even e.g. for problems that are more naturally solved by “working back from the goal position”?
  - For an example of such an environment, consider a graph with a root node $s$, which is always a starting node, a set $I$ of intermediate nodes, and a set $T$ of target nodes. Each target node is connected to a single intermediate node. Intermediate nodes can be connected among themselves arbitrarily. $s$ is connected all intermediate nodes. Given a target position $t$, one immediately knows the intermediate node it is connected to (because it is unique), and from there one can easily construct a path from $s$ to $t$. However, starting from $s$, there are $|I|$ possibilities for the first move. If $I$ is very large, it seems intuitive to me that solving the problem “backward” should be significantly easier given finite memory capacity compared to solving it “forward”. Please let me know if I’m missing something with this example!
- Do the authors expect there to be any major methodological hurdles in generalizing CompDiffuser to settings requiring processing of e.g. visual inputs? (or, more generally, non-ground-truth-state observations, POMDPs, etc).
- Recent work by Chen et al. (2024) introduced Diffusion Forcing, and had strong long rollout results extending to complex environments such as Minecraft. Do you see any complementarity in your approaches to e.g. enable stronger temporal generalization of decision making diffusion models in complex control settings? What would an ultimate, ideal solution for this more general problem in complex environments look like in your view, and what are the main unsolved problems for making it happen?
- Do the authors see a natural extension of their method to non-goal-directed settings, instead driven by e.g. reward functions or other success metrics?
- On a similar vein, would one expect CompDiffuser to inherit similar global guidance properties from full-trajectory diffusion methods like Diffuser or Decision Diffuser? My intuition is it would, since sub-plans condition on each other during sampling.

**Ethical Concerns:**

["NO or VERY MINOR ethics concerns only"]

**Final Justification:**

Following my observations in **Strengths** and the author's replies (particularly regarding ablations), I believe this work opens a lot of interesting directions for further work in compositional planning, and I maintain my recommendation of acceptance.

**Limitations:**

The authors address limitations in the conclusion.

**Paper Formatting Concerns:**

None.

**Quality:**

4

**Strengths And Weaknesses:**

**Strengths:**
- Clear problem motivation: the authors clearly outline their goal, which is to address length generalization in decision making diffusion models.
- Experimental results indicate successful completion of research goal and extensively study relevant variables in the method’s implementation and performance. In particular, they demonstrate that CompDiffuser does indeed show much better length generalization than its full-sequence diffusion counterparts.
- Thorough and high-quality exposition that makes it very clear what the proposed algorithmic approach is, and relevant design choices (e.g. sampling approach).

**Weaknesses:**
- Overall I consider this to be a good paper. On the algorithmic side, the three main novel points in their approach are: (1) subdividing large trajectories into chunks when denoising; (2) choosing the chunks to be overlapping; and (3) conditioning each chunk on its previous and following chunk. While ablations consider several aspects of the proposed setup (e.g. sensitivity to replanning, sampling approach, etc), I believe it would be constructive to include ablations of at least points (2) and (3).
  - Namely, these ablations would be: (2) testing a version of CompDiffuser where the chunks don’t overlap, and (3) testing a version of CompDiffuser that conditions only on the previous trajectory, and not on both adjacent ones. While I can see reasons why design decisions (2) and (3) should be beneficial, it’s not clear to me that e.g. the method would completely collapse or be mis-specified without them. These ablations would help understand the value of each algorithmic design decision made by the authors.
  - Also, ablation (3) above would bear some resemblance to Diffusion Forcing (Chen et al. 2023). It seems to me the paper would be strengthened by including a comparison between CompDiffuser and Diffusion Forcing in maze planning settings, since this is another recent diffusion decision making model which had good results relating to length generalization (e.g. infinite rollouts in Minecraft).

---

> ### Author Rebuttal · Authors · 2025-07-31
>
> We thank Reviewer yhYf for the positive and careful review. Please see our response to each listed concern below.
>
> > W1. It would be constructive to include ablations of at least points (2) and (3), where (2) is testing a version of CompDiffuser where the chunks don’t overlap and (3) is testing a version of CompDiffuser that conditions only on the previous trajectory.
>
>
> Thank you for your valuable suggestions.
>
> We include the ablation studies of inference with single-way conditioning below. As shown, the proposed bi-directional information propagation design outperforms the single-way conditioning counterpart that conditions only on the previous trajectory. The improvement becomes more prominent as the size of the maze increases, demonstrating the effectiveness of our method.
>
> | Env | Only Previous | Both (Ours) |
> | - | - | - |
> | Medium | 100±0   | 100±0 |
> | Large | 94±3  |  100±0 |
> | Giant | 17±3 |   68±3 |
>
>
> Due to limited time, we are unable to provide the numbers for the no-overlapping design by the deadline of the rebuttal -- we are running them now and will make sure they are in the final manuscript.
>
>
>
>
>
>
> > W2. It seems to me the paper would be strengthened by including a comparison between CompDiffuser and Diffusion Forcing in maze planning settings.
>
> Thank you for the valuable suggestions.
>
> We are working on adding this baseline, but due to limited time and compute, we are unable to provide the numbers for the no-overlapping design by the deadline of the rebuttal. We will make sure to include this baseline in the final manuscript.
>
>
>
>
>
> > Q1. Do the authors have a sense of, in the setting of Table 1, for how large a horizon does CompDiffuser’s performance start degrading, if any?
>
>
> Given that CompDiffuser achieves 100% in Large maze and 68% in Giant Maze, we speculate that CompDiffuser's success rate might start degrading if the planning horizon is between the horizon of the Large maze and Giant maze.
>
> Specifically, we compose 8 models to obtain a plan of horizon 888 for PointMaze-Giant-Stitch and 5 models for a plan of horizon 572 to solve PointMaze-Large-Stitch. Hence, for this specific setup, the CompDiffuser's performance might start degrading when composing 6-7 models, i.e., a horizon around 700.
>
>
>
> > Q2. do the authors have a sense of how they would build a CompDiffuser to achieve 100% accuracy in the Giant maze? Is it a matter of simply collecting more data? Or is it potentially a fundamental limitation of the method due to accumulating errors throughout the autoregressive sampling process at longer horizons?
>
>
>
> Thank you for your comments.
> We agree that the error accumulation when composing a large number of trajectories could negatively impact the performance. However, we believe that there are still multiple ways to enhance the performance of CompDiffuser.
> - We can leverage a plan verifier to select a promising plan out of a batch of candidate plans.
> - Inference-time searching [1] can be further integrated in the sampling process to enhance the performance.
> - More training data can also enhance the performance, because it can provide more intersection/overlapping between different trajectories, promoting cross-trajectory stitching. (The size of the Giant maze dataset is identical to the size of the datasets of Large maze and Medium Maze. However, since the Giant maze is much larger, its data density is actually smaller compared to the Large/Medium maze datasets.)
> - Increased replanning budget. Allowing more replanning steps may help the agent recover from failures, such as when getting stuck by walls.
>
> Please also see the response to Q3 below for additional strategies.
>
>
>
> > Q3. what are the main variables with respect to which the generalization ability of the method scales?
>
>
> Thank you for your insightful question. We believe that the following factors effect the generalization ability of the model:
> - Amount/Quality of data: We hypothesize that the quality of domain coverage and existence of overlaps between training trajectories will improve the performance of CompDiffuser, since higher state-space coverage and more intersections between trajectories in the dataset can promote cross-trajectories combinatorial generalization (hence encouraging stiching ability).
> - Ratio of training and eval trajectory length: Generally, we see that the performance starts to drop with increasing number of the composed trajectories. Hence, we believe that the performance of CompDiffuser can increase if it has access to longer training trajectories, as it will require less number of test-time composed trajectories.
> - Computation Budget: In our current setup, we use DDIM 50 steps to sample trajectories and we found that increasing the inference denoising timestep does not further improve the performance. However, we believe that techniques such as (1) inference-time searching technique [1] can be incorporated during the sampling process and (2) using an additional plan verifier to select a more reliable path will be helpful.
> - Replanning budget: Higher replanning budge can also enhance the performance of CompDiffuser. For instance, replanning can help the agent escape from suboptimal cases during a rollout (e.g., when the agent is stuck in a corner or collides into walls).
>
>
>
>
> > Q4. Do the authors expect the method to be robust to sampling direction in more general settings, even e.g. for problems that are more naturally solved by “working back from the goal position”?
>
>
>
> In the given example, we agree that sampling "backward" would be a more efficient sampling direction since it would be easier to find the intermediate node when starting from $t$ due to the high probability density (only one edge connecting to $t$).
> Though our method might still be able to solve such asymmetric case via its bi-directional information passing design (as in Figure 3), we believe that manually or leveraging an additional model to decide the denoising direction according to the task structure would be beneficial as this might reduce the planning search space.
>
>
>
>
>
> > Q5. Do the authors expect there to be any major methodological hurdles in generalizing CompDiffuser to settings requiring processing of e.g. visual inputs? (or, more generally, non-ground-truth-state observations, POMDPs, etc).
>
>
> Thank you for the valuable comments. As our method do not assume any specific knowledge about the tasks, it is generally applicable to any trajectory planning settings.
>
> For instance, if the starting observation is given in images, our method can use the latents of the images to condition the diffusion denoiser network to generate a trajectory of states.
> In addition, our method can also directly plan in the image-space (i.e., synthesize a sequence of images instead of states). A simple implementation is to use some existing pre-trained models such as Cosmos or DINO to encode the images into latents and run CompDiffuser on these latents.
>
>
> In the case of POMDPs, due to uncertainty about the underlying ground truth state of the environment, there may not be one correct ground truth plan we can follow to solve our desired task. In this setting,  we can leverage the probabilistic nature of CompDiffuser and sample multiple candidates trajectories, and then chose the sequence of actions that maximizes the likelihood of satisfying the set of the trajectories.
>
>
>
>
> > Q6. Complementarity between  Diffusion Forcing and the proposed method.
>
> Thank you for your insightful question! We believe that our method can be combined with the ideas in Diffusion Forcing to solve more complex long-horizon tasks. In such long-horizon tasks, it may not make sense to construct a fully detailed plan to solve the entire task, as such a plan can be computationally intractable to generate. We can instead adopt the idea of Diffusion Forcing, and learn each generative component with variable levels of noise. This then enables us to construct plans where individual portions of trajectories are either fully realized or abstracted, depending on the varied noise levels of each component, allowing us to more effectively generate long horizon plans
>
> Towards solving very complex, long-horizon problems -- we think that there are still open questions with respect to the right abstraction to plan over tasks (for instance, directly planning in continuous actions to make trip to Europe would not make sense), as well as how to form plans when the overall state of the world is unknown (for instance, making a plan to make dinner when you do not know the locations of pots or food).
>
>
>
> > Q7. Do the authors see a natural extension of their method to non-goal-directed settings, instead driven by e.g. reward functions or other success metrics?
>
>
> We believe that extending our method to non-goal-directed settings is an interesting future research topic. For example, a simple way to adapt our method to cost or reward directed planning is that when we are compositionally sampling $N$ segments, instead of using a goal state to guide the denoising of the final segment $\tau_N$, we can use some specific cost function or reward value as guidance.
> As we are not doing a final-goal directed planning, such guidance can also be flexibly applied to any intermediate segment $\tau_n$ to achieve other specific behavior.
>
>
>
> > Q8. Would one expect CompDiffuser to inherit similar global guidance properties from full-trajectory diffusion methods like Diffuser or Decision Diffuser?
>
>
> We believe that CompDiffuser is able to inherit the global guidance properties from full-trajectory diffusion methods.
>
> Though we are sampling multiple segments $\tau_{1:N}$ concurrently as in Figure 3, each segment actually depends on the values of other segments. Hence, the sampling of the segment set $\tau_{1:N}$ can be viewed as a holistic sampling process, which enables the global guidance properties.
>
>
>
>
> [1] Inference-time Scaling of Diffusion Models through Classical Search

---

> > ### Comment · Reviewer_yhYf · 2025-08-04
> > **Thank you for the extensive reply!**
> >
> > Thank you for carefully replying to my questions about CompDiffuser, especially relating to ablations, how one can boost CompDiffuser's performance, and potential algorithmic improvements by leveraging recent techniques from the literature.
> >
> > I believe this work opens a lot of interesting directions for further work in compositional planning, and I maintain my recommendation of acceptance.

---

> > > ### Author Response · Authors · 2025-08-05
> > > **Thank you**
> > >
> > > Dear Reviewer yhYf,
> > >
> > > We sincerely appreciate your thoughtful and detailed feedback, as well as your positive evaluation of our manuscript. Thank you once again for your time and effort in reviewing our work.
> > >
> > > Best,
> > > Paper Authors

---

### Note · Authors · 2025-08-15

We sincerely thank all reviewers and ACs for their time, thoughtful feedback, and positive evaluations of our work.


In addition to addressing the per-review questions, we conducted new experiments during the rebuttal period, including: evaluation in a robotic manipulation environment, ablation studies of single-way conditioning at inference, comparison with Decision Diffuser baseline with matched model size, and additional experiments on joint planning over both qpos and qvel.

Below, we provide additional results for the experiments suggested by Reviewer yhYf.

We evaluate an additional baseline, Diffusion Forcing, on the OGBench PointMaze Stitch datasets. We use its official codebase with default hyperparameters, except that we set the training horizon to 160 to match our method. Qualitatively, we observe that in test time Diffusion Forcing struggles to generate consistent plans for long-horizon goals. One potential reason is its guidance term: Diffusion Forcing directly uses the gradient of the L2-distance between the goal and the denoised plan as guidance throughout the denoising process, which simply pulls most intermediate states towards the goal and yields fragmented trajectories. Furthermore, Diffusion Forcing might also be more susceptible to the accumulated errors over long rollouts, resulting in such incoherent plans or trajectories that pass through walls.


| Env | Diffusion Forcing | Ours |
| - | - | - |
| Medium | 33±0   | 100±0 |
| Large | 60±0  |  100±0 |
| Giant | 0±0 |   68±3 |


We also present an ablation study of the segment-overlap design on the OGBench PointMaze Stitch datasets. We train a variant where the segment being denoised does not overlap with its neighbor segments. At test time, we still use the same sampling method as Algorithm 2. After diffusion sampling, to construct the final trajectory $\tau_{comp}$, because the segments $x_{1:N}$ do not have any overlaps with each other, we directly concatenate all these segments instead of applying trajectory merging. As shown in the table below, our method is still effective when there are no overlaps between segments, while the proposed overlapping design can further enhance performance and yield additional gains in success rate.

| Env | w/o Overlap | w/ Overlap (Ours) |
| - | - | - |
| Medium |  100±0 | 100±0 |
| Large | 98±1  |  100±0 |
| Giant | 61±5 |   68±3 |

---

### Decision · Program_Chairs · 2025-09-17

**Decision:**

Accept (spotlight)

**Comment:**

This paper introduces CompDiffuser, a novel diffusion-based composition method for generative planning. The paper is well-written, addresses an important problem, and presents a method that is clearly motivated and supported by extensive and successful results. It received unanimous positive reviews. During the rebuttal and discussion phases, the authors effectively addressed reviewers’ questions by providing new experimental results in a robotic manipulation environment, additional ablation studies on single-way conditioning at inference, a fair comparison with the Decision Diffuser baseline using matched model sizes, and further experiments on joint planning over both qpos and qvel. Overall, this is a strong and interesting paper that merits acceptance.